# Dynamic Neighborhood Construction for Structured Large Discrete Action Spaces

**Fabian Akkerman**[*,1,†]**, Julius Luy**[*,2,†]**, Wouter van Heeswijk**[1]**, Maximilian Schiffer**[2,3]

[1] University of Twente (7500 AE Enschede, The Netherlands)
[2] School of Management, Technical University of Munich (80333 Munich, Germany)
[3] Munich Data Science Institute, Technical University of Munich (85748 Garching, Germany)

## Abstract

Large discrete action spaces (LDAS) remain a central challenge in reinforcement learning. Existing solution approaches can handle unstructured LDAS with up to a few million actions. However, many real-world applications in logistics, production, and transportation systems have combinatorial action spaces, whose size grows well beyond millions of actions, even on small instances. Fortunately, such action spaces exhibit structure, e.g., equally spaced discrete resource units. With this work, we focus on handling structured LDAS (SLDAS) with sizes that cannot be handled by current benchmarks: we propose Dynamic Neighborhood Construction (DNC), a novel exploitation paradigm for SLDAS. We present a scalable neighborhood exploration heuristic that utilizes this paradigm and efficiently explores the discrete neighborhood around the continuous proxy action in structured action spaces with up to $10^{73}$ actions. We demonstrate the performance of our method by benchmarking it against three state-of-the-art approaches designed for large discrete action spaces across three distinct environments. Our results show that DNC matches or outperforms state-of-the-art approaches while being computationally more efficient. Furthermore, our method scales to action spaces that so far remained computationally intractable for existing methodologies.

## 1 Introduction

In deep reinforcement learning (DRL), ample methods exist to successfully handle large state spaces, but methods to handle large discrete action spaces (LDAS) remain scarce (Dulac-Arnold et al., 2021). Off-the-shelf DRL algorithms – e.g., Deep Q-Networks (DQN) (Mnih et al., 2013), Deep Policy Gradients (DPG) (Silver et al., 2014), or Proximal Policy Optimization (PPO) (Schulman et al., 2017) – fail to handle such LDAS, as they require in- or output nodes for each discrete action, which renders learning accurate $Q$-values or action probabilities computationally intractable. Recent methods extending off-the-shelf algorithms focus on applications with unstructured LDAS, found in, e.g., recommender systems or autonomous tool selection (cf. Dulac-Arnold et al., 2015; Chandak et al., 2019a; Jain et al., 2020). Herein, the agent's action space encodes an irregular set of features. To handle LDAS, current DRL research suggests to learn a continuous policy and utilize an output mapping to each discrete action (Dulac-Arnold et al., 2015; Chandak et al., 2019a), which allows solving problems with moderately large action spaces up to a few million actions. However, many real-world problems have (combinatorial) action spaces exhibiting billions of actions even for small problem instances. Examples are the management of order levels for items in a large warehouse (Boute et al., 2022) and the synthesis of new chemical structures for drug design (Gottipati et al., 2020). Current state-of-the-art techniques for LDAS cannot scale to such very large LDAS as they define an explicit mapping for each discrete action.

Yet, many problems with combinatorial action spaces exhibit structure, e.g., equally spaced discrete resource units in inventory replenishment (Boute et al., 2022), distance units in path planning for autonomous vehicles (Xie et al., 2023), the movement of robot joints in a discretized action space (Neunert et al., 2020), and traffic signal control ordered by the sequence of lanes (Li et al., 2021). With this work, we show how to exploit such structure, eliminating the need to enumerate all actions, hence

---

[*]Equal contribution.
[†]Corresponding authors: Fabian Akkerman (f.r.akkerman@utwente.nl), Julius Luy (julius.luy@tum.de).

allowing to scale to much larger action spaces. Specifically, we propose a novel algorithmic pipeline that focuses on structured LDAS (SLDAS) and exploits their structure by embedding continuous-to-discrete action mappings via dynamic neighborhood construction into an actor-critic algorithm. This pipeline overcomes the scalability limitations of the state of the art on SLDAS by bypassing explicit action space enumeration, scaling up to action spaces of size $10^{73}$, while showing comparable or even improved algorithmic performance.

**Related Literature**  To handle LDAS, some works employ factorization to reduce the action space's size by grouping actions into representations that are easier to learn. Examples are factorization into binary subsets (Sallans & Hinton, 2004; Pazis & Parr, 2011; Dulac-Arnold et al., 2012), expert demonstrations (Tennenholtz & Mannor, 2019), tensor factorization (Mahajan et al., 2021), symbolic representations of state-action values (Cui & Khardon, 2016; 2018), or the use of value-based DRL, incorporating the action space structure into a $Q$-network (Tavakoli et al., 2018; Sharma et al., 2017; Wei et al., 2020; Tang et al., 2022). Hierarchical reinforcement learning (HRL) and multi-agent reinforcement learning (MARL) approaches effectively employ factorization as well (Zhang et al., 2020b; Kim et al., 2021; Peng et al., 2021; Enders et al., 2023). Although some of these works cover extremely large action spaces and prove to be effective for specific problem classes, the proposed approaches require an a priori encoding for each discrete action, do not leverage the action space's structure, and frequently require substantial design- and parameter tuning effort (particularly in the case of MARL).

While factorization methods reduce the number of evaluated actions, methods that leverage continuous-to-discrete mappings consider the continuum between discrete actions, converting continuous actions to discrete ones subsequently. van Hasselt & Wiering (2009) use an actor-critic algorithm, rounding the actor's continuous output to the closest integer to obtain a discrete counterpart. Vanvuchelen et al. (2022) extend this concept to multi-dimensional action vectors, normalizing the actor's output and rounding it to the nearest discrete value. Such rounding techniques are straightforward and computationally efficient, but may be unstable if their mapping is too coarse. Different continuous outputs might be mapped to the same discrete action, while ignoring potentially better neighbors. To mitigate this issue, Dulac-Arnold et al. (2015) and Wang et al. (2021) replace the rounding step by a $k$-nearest neighbor search or $k$-dimensional tree search. While such methods allow handling both unstructured- and SLDAS, they offer only limited scalability, necessitating the definition and storage of the discrete action space a priori, e.g., in matrix form.

Other works aim at learning action representations. Thomas & Barto (2012) use a goal-conditioned policy to learn motor primitives, i.e., aggregated abstractions of lower-level actions. Chandak et al. (2019a) consider a policy that maps continuous actions into an embedding space and employ supervised learning to identify a unique embedding for each discrete action. Follow-up works consider combined state-action embeddings (Whitney et al., 2020; Pritz et al., 2021), or reduce the impact of out-of-distribution actions in offline DRL by measuring behavioral and data-distributional relations between discrete actions (Gu et al., 2022). Some works propose to learn embeddings for all feasible actions by means of a value-based approach (He et al., 2016), or learn to predict and avoid suboptimal actions (Zahavy et al., 2018). Chandak et al. (2019b) and Jain et al. (2020; 2022) consider the problem of adapting an RL agent to previously unseen tasks by mapping the structure of unseen actions to the set of already observed actions. In general, action representation learning allows to handle unstructured and SLDAS, but requires a vast amount of data to learn accurate representations, often hampering algorithmic performance. Moreover, scalability issues remain due to learning dedicated representations for each action.

As can be seen, existing methods to handle SLDAS are constrained by the action space size due the following obstacles: (i) straightforward factorization approaches require handcrafted encodings a priori and are thus confined to enumerable action spaces, (ii) approaches that base on HRL or MARL are highly problem-specific and rely on intense hyperparameter tuning, (iii) static mappings lack scalability as they require to define—and store—the discrete action space a priori, (iv) learning action representations requires a vast amount of data to obtain dedicated representations for each action. Thus, all existing approaches are upper-bounded on the number of actions they can handle.

**Contribution**  To close the research gap outlined above, we propose a novel algorithmic pipeline that exploits the structure in SLDAS while maintaining or improving the state-of-the-art in solution quality and reaching a new level of scalability to very large SLDAS. This pipeline embeds continuous-

to-discrete action mappings via dynamic neighborhood construction (DNC) into an actor-critic algorithm. DNC is leveraged to convert the actor's continuous output into a discrete action via a simulated annealing (SA) search, guided by discrete actions' $Q$-values derived from its critic. Our approach has two novel components: (i) we exploit the structure of the action space using dynamic neighborhood construction, and (ii) we employ an SA neighborhood search to improve performance. Although our approach classifies as a continuous-to-discrete action mapping, it requires no a priori definition of the action space. Moreover, our approach can be applied to a wide range of problem settings as long as the respective action space exhibits structure. We benchmark our pipeline against three state-of-the-art approaches (Dulac-Arnold et al., 2015; Chandak et al., 2019a; Vanvuchelen et al., 2022) and a vanilla actor critic (VAC) baseline across three environments: a maze environment, a joint inventory replenishment, and a job-shop scheduling problem. Our results verify the performance of our pipeline: it scales up to discrete action spaces of size $10^{73}$, while showing comparable or improving solution quality across all investigated environments.

## 2 PROBLEM DESCRIPTION

We study discrete, sequential decision-making problems formalized as Markov decision processes (MDPs), described by a state space $\mathcal{S}$, a discrete action space $\mathcal{A}$, a reward function $r : \mathcal{S} \times \mathcal{A} \to \mathbb{R}$, and transition dynamics $\mathbb{P} : \mathcal{S} \times \mathcal{A} \times \mathcal{S} \to [0, 1]$. We represent states $\boldsymbol{s} \in \mathcal{S}$ and actions $\boldsymbol{a} \in \mathcal{A}$ by $N$-and $M$-dimensional vectors, such that $\boldsymbol{s} \in \mathbb{R}^M$ and $\boldsymbol{a} \in \mathbb{N}^N$. We consider multi-dimensional action vectors to ensure general applicability, but refer to such a vector as an action in the remainder of this paper for the sake of conciseness. We focus on regularly structured action spaces, which we can represent with a regular tessellation of an $n$-dimensional hypercube or hyperrectangle. Hence, we assume that the smallest distance between two neighboring vertices of the tesselated grid along each dimension $n$, $\Delta a_n$, is constant. Figures 1a and 1b visualize two examples of regularly structured grids, that can be found in, e.g., inventory problems, wherein coordinates $a_1$ and $a_2$ represent the discrete resource units of items 1 and 2 respectively (cf. Boute et al., 2022). Figure 1c exemplifies an irregularly structured action space. Note that our method extends to $\boldsymbol{a} \in \mathbb{Q}^N$ as long as the action space can be represented by a regular grid. For conciseness we will refer to "regularly structured" as "structured" action spaces and to "irregularly structured" as "unstructured" action spaces in the remainder.

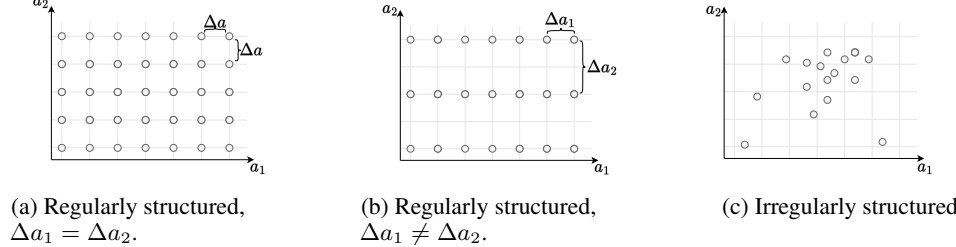

(a) Regularly structured, $\Delta a_1 = \Delta a_2$.

(b) Regularly structured, $\Delta a_1 \neq \Delta a_2$.

(c) Irregularly structured.

Figure 1: Structured and unstructured action spaces in 2D. Each vertex represents an action.

Let us denote a policy by $\pi : \mathcal{S} \to \mathcal{A}$ and the state-action value function by $Q^\pi(\boldsymbol{s}, \boldsymbol{a}) = \mathbb{E}_\pi \left[ \sum_{t=0}^\infty \gamma^t r_t | \boldsymbol{s}, \boldsymbol{a} \right]$, where $\gamma \in [0, 1)$ denotes the discount factor. We aim to find a policy that maximizes the objective function $J = \mathbb{E}_{\boldsymbol{a} \sim \pi}[Q^\pi(\boldsymbol{s}, \boldsymbol{a})]$.

To introduce our method, first consider an actor-critic framework, in which an actor determines action $\boldsymbol{a} \in \mathcal{A}$ based on a policy (actor network) $\pi_{\boldsymbol{\theta}}$ parameterized by weight vector $\boldsymbol{\theta}$, and a critic parameterized by $\boldsymbol{w}$ which estimates the value of this action, i.e., returns $Q_{\boldsymbol{w}}(\boldsymbol{s}, \boldsymbol{a})$. The actor is updated in the direction suggested by the critic by maximizing the objective function $J$. The actor network's output layer contains $|\mathcal{A}|$ nodes, each reflecting the probability of an action $\boldsymbol{a} \in \mathcal{A}$, i.e., the network $\pi_{\boldsymbol{\theta}}(\boldsymbol{s})$ encodes the policy. In many real-world problems, $|\mathcal{A}|$ grows exponentially with the state dimension. In these cases, obtaining an accurate policy requires vast amounts of training data and exploration to ensure generalization over $\mathcal{A}$, making training of $\pi_{\boldsymbol{\theta}}$ intractable.

To bypass the need of learning accurate probabilities for each action, we will solve the following surrogate problem in the remainder of this paper: instead of finding a discrete policy returning action

probabilities for each $\boldsymbol{a} \in \mathcal{A}$, we find a policy that returns continuous actions $\hat{\boldsymbol{a}} \in \mathbb{R}^N$ and a function $f(\hat{\boldsymbol{a}}) = \boldsymbol{a}$ that maps continuous actions to discrete ones. This approach allows to use off-the-shelf policy gradient algorithms – which perform well in continuous action spaces – to learn $\pi_{\boldsymbol{\theta}}(\boldsymbol{s})$ and ensures that $\pi_{\boldsymbol{\theta}}$'s output layer grows linearly with the number of entries in $\hat{\boldsymbol{a}}$.

A straightforward implementation of $f$ would return the discrete neighboring action to $\hat{\boldsymbol{a}}$ with the highest $Q$-value, as proposed in Dulac-Arnold et al. (2015). Herein, the authors search for the $k$ closest neighbors to $\hat{\boldsymbol{a}}$ over the entire action space and return the discrete action with the highest $Q$-value. This requires an *a priori* definition of the action space, e.g., encoded as a matrix, hence reaching memory limits if action spaces are very large. Instead of searching the entire action space, we aim to efficiently search for an action $\boldsymbol{a}$ that maximizes $Q$. Formally, one can describe the underlying optimization problem as a mixed-integer linear program (MILP); the action space structure and the actions' discreteness allow imposing linear constraints to bound the action vector's entries and, hence, for an efficient neighborhood search. $Q$-values are represented by deep neural networks (DNNs) to capture non-linearities, which we can express as a MILP if the DNN solely consists of linear and ReLU activation functions (cf. Bunel et al., 2017). Let $l \in \mathcal{L} = \{1, \dots, L\}$ be the index of the $l$-th DNN layer and, with slight abuse of notation, $d_l$ denote neuron $d \in \mathcal{D}_l = \{1, \dots, D_l\}$ in layer $l$. Let $o$ be the DNN's output node $o$ and $w_{d_{l-1}, d_l}$ be the DNN's weights. The outputs $y_l$ of the DNN's neurons and the discrete action vector elements $a_i \in \boldsymbol{a}$, $i \in \{1, \dots, N\}$, represent the MILP's decision variables. As we assume the discrete action space has an underlying continuous structure, we expect the best discrete action to be close in vector distance to the continuous proxy action and restrict our search space accordingly. Let $a_i^{u'}(\hat{\boldsymbol{a}})$ and $a_i^{l'}(\hat{\boldsymbol{a}})$ denote upper and lower bounds on the action vector's elements respectively. The MILP with the main set of constraints then reads:

$$\max_{\boldsymbol{a}} \left( Q_{\boldsymbol{w}}(\boldsymbol{s}, \boldsymbol{a}) \right) = \max_{a_i, y_{d_l}} \sum_{d_L} w_{d_L, o} \, y_{d_L} \tag{1}$$

$$\text{s.t.} \quad a_i^{l'}(\hat{\boldsymbol{a}}) \leq a_i \leq a_i^{u'}(\hat{\boldsymbol{a}}), \qquad\qquad \forall i \in \{1, \dots, N\}, \tag{2}$$

$$y_{d_l} \geq 0, \qquad\qquad \forall l \in \mathcal{L}, d \in \mathcal{D}, \tag{3}$$

$$y_{d_l} \geq \sum_{d_{l-1}=1}^{|\boldsymbol{a}| < D_l} w_{d_{l-1}, d_l} \, a_{d_{l-1}} + \sum_{d_{l-1}=|\boldsymbol{a}|}^{D_l} w_{d_{l-1}, d_l} \, y_{d_{l-1}}^{\boldsymbol{s}}, \qquad l = 2, \forall d_l. \tag{4}$$

The Objective (1) maximizes the weighted sum of outputs in the last layer, which equals $Q_{\boldsymbol{w}}(\boldsymbol{s}, \boldsymbol{a})$. Constraints 2 describe the neighborhood of $\hat{\boldsymbol{a}}$. Constraints 3 implement the ReLU activation functions by imposing non-negativity on the neurons' outputs. Constraints 4 describe the transition from the input to the second DNN layer. Here, $y_{d_{l-1}}^{\boldsymbol{s}}$ are the outputs of previous feedforward layers processing only the state input $\boldsymbol{s}$. We provide the full set of constraints in the supplement. The MILP bypasses the need to store the entire action space matrix in memory, but needs to be solved in each training step as the DNN weights change, which is costly for large DNN architectures. Accordingly, we base our methodology in the following on efficiently finding a heuristic solution to this MILP via dynamically constructing and searching through local neighborhoods of $\hat{\boldsymbol{a}}$.

## 3 METHODOLOGY

Figure 2 shows the rationale of our algorithm's pipeline, which builds upon an actor-critic framework, leveraging DNC to transform the actor's continuous output into a discrete action. Our pipeline comprises three steps. First, we use the actor's output $\hat{\boldsymbol{a}}$ to generate a discrete base action $\bar{\boldsymbol{a}} \in \mathcal{A}$. We then iterate between generating promising sets of discrete neighbors $\mathcal{A}'$, and evaluating those based on the respective $Q$-values taken from the critic. Here, we exploit the concept of SA (cf. Kochenderfer & Wheeler, 2019) to guide our search and ensure sufficient local exploration. The remainder of this section details each step of our DNC procedure and discusses our algorithmic design decisions.

**Generating a Discrete Base Action** We consider an actor network whose output corresponds to the first-order $\mu_{\boldsymbol{\theta}}(\boldsymbol{s})_n \in \mathbb{R}$ and second-order $\sigma_{\boldsymbol{\theta}}(\boldsymbol{s})_n \in \mathbb{R}$ moments of pre-specified distributions for each element $n \in \{1, \dots, N\}$ of the action vector to parameterize a stochastic policy $\pi_{\boldsymbol{\theta}}(\boldsymbol{s})$ with continuous actions $\hat{\boldsymbol{a}}$. To obtain discrete base actions, we scale and round each continuous $\hat{a}_i$ to the closest discrete value, denoting the corresponding base action after rounding by $\bar{\boldsymbol{a}}$. We refer to the supplementary material for further details on this generation procedure.

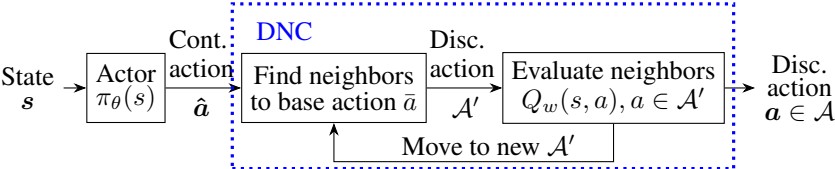

Figure 2: Pipeline for finding discrete actions in SLDAS

**Generating Sets of Discrete Neighbors** Within DNC, we aim to leverage neighbors of discrete base actions, motivated by the rationale that neighborhoods of actions exhibit a certain degree of cohesion. Specifically, when generating action neighborhoods $\mathcal{A}'$, we premise that action pairs with small vector distances generate similar $Q$-values.

**Definition 1** *We measure action similarity within $\mathcal{A}'$ via a Lipschitz constant $L$ satisfying $|Q^\pi(s, a) - Q^\pi(s, a')| \leq L\|a - a'\|_2$ for all $a, a' \in \mathcal{A}'$.*

**Lemma 1** *Action similarity $L$ is given by $\displaystyle\sup_{a, a' \in \mathcal{A}', a \neq a'} \frac{|Q^\pi(s, a) - Q^\pi(s, a')|}{\|a - a'\|_2}$, ensuring that $|Q^\pi(s, a) - Q^\pi(s, a')| \leq L\|a - a'\|_2$ for all $a, a' \in \mathcal{A}'$.*

To prove Lemma 1, we use that $\mathcal{A}'$ is discrete, finite, and maps onto the real domain, and thus a finite $L$ exists. We refer to the supplementary material for a formal proof.

To generate discrete neighbors of $\bar{a}$, we perturb each action vector entry $\bar{a}_n$, $n \in \{1, \ldots, N\}$. To do so, we define a perturbation matrix $\boldsymbol{P} = (P_{ij})_{i=1,\ldots,N;j=1,\ldots,2dN}$, with $d$ being the neighborhood depth. Moreover, let $\epsilon$ denote a scaling constant that allows us to look at more distant (larger $\epsilon$) or closer (smaller $\epsilon$) neighbors. With this notation, the perturbation matrix $\boldsymbol{P}$ reads as follows:

$$P_{ij} = \begin{cases} \epsilon\left(\lfloor(j-1)/N\rfloor + 1\right), & \text{if, } j \in \{i, i+N, i+2N, \ldots, i+(d-1)N\}, \\ -\epsilon\left(\lfloor(j-1)/N\rfloor + 1 - d\right), & \text{if, } j \in \{i+dN, i+(d+1)N, \ldots, i+(2d-1)N\}, \\ 0, & \text{otherwise.} \end{cases}$$

The first $d \cdot N$ columns of $\boldsymbol{P}$ are vectors with one non-zero entry describing a positive perturbation of each entry in $\hat{a}$, the last $d \cdot N$ columns describe negative perturbations. Let $\bar{\boldsymbol{A}} = (\bar{A}_{ij})_{i=1,\ldots,N;j=1,\ldots,2dN}$ denote a matrix that stores $\bar{a}$ in each of its columns. We then obtain the perturbed matrix $\boldsymbol{A} = (A_{ij})_{i=1,\ldots,N;j=1,\ldots,2dN}$ as $\boldsymbol{A} = \bar{\boldsymbol{A}} + \boldsymbol{P}$, such that $\boldsymbol{A}$'s columns form the set $\mathcal{A}'$, yielding $2 \cdot N \cdot d$ neighbors with maximum $L_2$ distance $(d\,\epsilon)$. In each perturbation, we modify only one entry of $\bar{a}$, such that each perturbed neighbor has a Hamming distance of 1 and an $L_2$ distance of up to $(d\,\epsilon)$ to the base action. Note that we can define $\epsilon$ individually for each dimension $n$, which we omit in this description to remain concise.

The perturbation approach allows for efficient implementation and scales to very large action spaces by design as it limits the exploration of the neighborhood of $\bar{a}$, which in a general case would still increase exponentially with respect to a maximum $L_2$-distance of $(d\,\epsilon)$. Clearly, limiting the neighborhood evaluation may incur a performance loss, but we argue that this loss is limited, as our search procedure described in the next section is able to recover initially ignored neighbors.

Assuming that an action's structured neighborhood relates to a locally convex $J$ w.r.t. action vector changes, we can show that the worst-case performance relative to base action $\bar{a}$ is bounded within a radius corresponding to the maximum perturbation distance $(d\,\epsilon)$. To formalize this, let $\mathcal{A}'' = \{a \in \mathcal{A}' : \|a - \bar{a}\|_2 = (d\,\epsilon)\}$ denote the set of maximally perturbed actions with respect to $\bar{a}$.

**Lemma 2** *If $J$ is locally upward convex for neighborhood $\mathcal{A}'$ with maximum perturbation $(d\,\epsilon)$ around base action $\bar{a}$, then worst-case performance with respect to $\bar{a}$ is bound by the maximally perturbed actions $a'' \in \mathcal{A}''$ via $Q^\pi(s, a') \geq \displaystyle\min_{a'' \in \mathcal{A}''} Q^\pi(s, a''), \forall a' \in \mathcal{A}'$.*

To prove Lemma 2 we leverage that inequality $\displaystyle\min_{a'' \in \mathcal{A}''} Q^\pi(s, a'') \leq Q^\pi(s, \lambda(a) + (1 - \lambda)(a'))$, with $a, a' \in \mathcal{A}''$ and $\lambda \in [0, 1]$, holds by definition of upward convexity and refer to the supplementary material for a complete proof.

**Evaluating Discrete Action Neighborhoods**    Algorithm 1 details the evaluation of a base action's neighborhood and our final selection of the discrete action $\boldsymbol{a}$ that we map to $\hat{\boldsymbol{a}}$. We initially select the discrete action within a neighborhood that yields the highest $Q$-value. However, our selection does not base on a single neighborhood evaluation, but employs an iterative SA-based search scheme to explore various neighborhoods. SA is a probabilistic search technique that facilitates to escape local optima during a search by occasionally accepting worse actions than the best one found (cf. Kochenderfer & Wheeler, 2019).

Specifically, Algorithm 1 works as follows. After generating a set of neighbors $\mathcal{A}'$ (l.3), we utilize the critic to obtain $Q$-values for $\boldsymbol{a}' \in \mathcal{A}'$, which includes the current base action $\bar{\boldsymbol{a}}$ and each neighbor (l.4). Subsequently, we store the $k$-best neighbors in an ordered set $\mathcal{K}' \subseteq \mathcal{A}'$ and store $\mathcal{K}'$ in $\mathcal{K}$, the latter set memorizing all evaluated actions thus far (l.5). From $\mathcal{K}'$, we select action $\boldsymbol{k}_1$, which by definition of $\mathcal{K}'$ has the highest associated $Q$-value, for evaluation (l.6). If the $Q$-value of $\boldsymbol{k}_1 \in \mathcal{K}'$ exceeds the $Q$-value of the base action $\bar{\boldsymbol{a}}$, we accept $\boldsymbol{k}_1$ as new base action $\bar{\boldsymbol{a}}$ (l.8). If it also exceeds the current best candidate action, $\bar{\boldsymbol{a}}^*$ (l.10), we accept it as the new best candidate action. If the action $\boldsymbol{k}_1$ does not exhibit a higher $Q$-value than $\bar{\boldsymbol{a}}$, we accept it with probability $1 - \exp[-\left(Q_w(\bar{\boldsymbol{a}}) - Q_w(\boldsymbol{k}_1)\right)/\beta]$ and reduce $\beta$ by cooling parameter $c_\beta$ (l.12), or reject it and move to a new base action, sampled from $\mathcal{K}$ (l.14). Finally, the parameter $k$ is reduced by cooling parameter $c_k$. Steps (l.3) to (l.15) are repeated until the stopping criterion (l.2) has been met. We then set $\boldsymbol{a} = \bar{\boldsymbol{a}}^*$, i.e., we use the action with the highest $Q$-value found as our final discrete action. For a more detailed description of the search process, hyperparameter tuning, and the integration of Algorithm 1 into a generic actor-critic algorithm, we refer to the supplementary material.

---

**Algorithm 1** Dynamic Neighborhood Construction

---

1:  Initialize $k, \beta, c_\beta$, and $c_k$, $\bar{\boldsymbol{a}} \leftarrow g(\hat{\boldsymbol{a}}), \bar{\boldsymbol{a}}^* \leftarrow \bar{\boldsymbol{a}}, \mathcal{K} = \emptyset$
2:  **while** $k > 0$, or $\beta > 0$ **do**
3:      Find neighbors $\mathcal{A}'$ to $\bar{\boldsymbol{a}}$ with $P_{ij}$
4:      Get $Q$-values for all neighbors in $\mathcal{A}'$
5:      Obtain set $\mathcal{K}'$ with $k$-best neighbors, $\mathcal{K} \leftarrow \mathcal{K} \cup \mathcal{K}'$
6:      $\boldsymbol{k}_1 \leftarrow \mathcal{K}'$, with $\boldsymbol{k}_1 = \arg\max_{k \in \mathcal{K}'} Q_w(\boldsymbol{s}, \boldsymbol{k})$
7:      **if** $Q_w(\boldsymbol{s}, \boldsymbol{k}_1) > Q_w(\boldsymbol{s}, \bar{\boldsymbol{a}})$ **then**
8:          Accept $\boldsymbol{k}_1 \in \mathcal{K}'$, $\bar{\boldsymbol{a}} \leftarrow \boldsymbol{k}_1$
9:          **if** $Q_w(\boldsymbol{s}, \boldsymbol{k}_1) > Q_w(\boldsymbol{s}, \bar{\boldsymbol{a}}^*)$ **then**
10:             $\bar{\boldsymbol{a}}^* \leftarrow \boldsymbol{k}_1$
11:     **else if** rand() $< \exp[-\left(Q_w(\bar{\boldsymbol{a}}) - Q_w(\boldsymbol{k}_1)\right)/\beta]$ **then**
12:         Accept $\boldsymbol{k}_1 \in \mathcal{K}'$, $\bar{\boldsymbol{a}} \leftarrow \boldsymbol{k}_1$, $\beta \leftarrow \beta - c_\beta$
13:     **else**
14:         Reject $\boldsymbol{k}_1 \in \mathcal{K}'$, $\bar{\boldsymbol{a}} \leftarrow \boldsymbol{k}_{\text{rand}} \in \mathcal{K}$
15:     $k \leftarrow \lceil k - c_k \rceil$
16:  Return $\bar{\boldsymbol{a}}^*$

---

**Discussion**    A few technical comments on the design of our algorithmic pipeline are in order. First, a composed policy of the form $\boldsymbol{a} = \pi'_{\boldsymbol{\theta}}(\boldsymbol{s}) = \text{DNC}(\pi_{\boldsymbol{\theta}}(\text{s}))$ is not fully differentiable. To ensure that our overall policy's backpropagation works approximately, we follow two steps, similar to Dulac-Arnold et al. (2015). Step 1 bases the actor's loss function on the continuous action $\hat{\boldsymbol{a}}$ to exploit information from the continuous action space that would have been lost if training the actor on the discrete action $\hat{\boldsymbol{a}}$. Step 2 trains the critic using the discrete action $\boldsymbol{a}$, i.e., the actions that were applied to the environment and upon which rewards were observed. We detail the integration of DNC into an actor-critic algorithm in the supplementary material. Second, one may argue that using an iterative SA-based algorithm is superfluous, as our overall algorithmic pipeline already utilizes a stochastic policy $\pi_{\boldsymbol{\theta}}$ that ensures exploration. Here, we note that utilizing $\pi_{\boldsymbol{\theta}}$ only ensures exploration in the continuous action space. To ensure that subsequent deterministic steps in $\mathcal{A}$, i.e., base action generation and neighborhood selection, do not lead to local optima, we use SA to search across different and potentially better neighborhoods in the discrete action space. Furthermore, as DNC does not explore the neighborhood as exhaustively as, e.g., a $k$nn or a MILP based approach, SA ensures that potentially better neighbors are not left out. In fact, we can show that with the chosen algorithmic design, DNC leads to improving actions in finite time.

**Lemma 3** *Consider a neighborhood $\mathcal{A}'$ and improving actions satisfying $Q^\pi(\boldsymbol{s}, \boldsymbol{a}) > \max_{\boldsymbol{a}' \in \mathcal{A}'} Q^\pi(\boldsymbol{s}, \boldsymbol{a}'), \boldsymbol{a} \in \mathcal{A} \setminus \mathcal{A}'$. In finite time, DNC will accept improving actions, provided that (i) $\beta$ and $k$ cool sufficiently slowly and (ii) a maximum perturbation distance $(d \, \epsilon)$ is set such that all action pairs can communicate.*

To prove Lemma 3, we utilize that under conditions (i) and (ii), our SA-based search can be formalized as an irreducible and aperiodic Markov chain over $\mathcal{A}$. A positive transition probability then applies to each action pair. For a complete proof, we refer to the supplementary material.

## 4 EXPERIMENTAL DESIGN

We compare the performance of our algorithmic pipeline (DNC) against four benchmarks: a VAC, the static MinMax mapping proposed in Vanvuchelen et al. (2022), the $k$-nearest neighbors ($k$nn) mapping proposed in Dulac-Arnold et al. (2015), and the learned action representation (LAR) approach proposed in Chandak et al. (2019a). To this end, VAC can be seen as a baseline, while MinMax, knn, and LAR denote state-of-the-art benchmarks. Furthermore, to show the added value of SA, we conduct an ablation study in which SA is removed from the solution method, which we label with DNC w/o SA. Here, we generate a set of discrete neighbors once and return the best action directly. We detail all of these benchmarks and respective hyperparameter tuning in the supplementary material. We consider three different environments to analyze algorithmic performance, which we summarize in the following. For their implementation details, we refer to the supplementary material.

First, we consider a maze environment, which is used as a testbed in state-of-the-art works on LDAS (cf. Chandak et al., 2019a). Here, an agent needs to navigate through a maze, avoiding obstructions and finding a goal position. The agent receives continuous coordinates on its location as input and decides on the activation of $N$ actuators– equally spaced around the agent– that move the agent in the direction they are pointing at. The resulting action space is exponential in the number of actuators, i.e., $|\mathcal{A}| = 2^N$. The agent incurs a small negative reward for each step and a reward of $100$ when the goal is reached. Random noise of $10\%$ is added to every action. In addition, we consider a variation of the maze problem, wherein we add one teleporting tile that needs to be avoided when trying to find the goal. Landing on this tile will result in a penalty of -20 and teleport the agent back to the starting position. With this variant, we study the policies' ability to deal with a non-linear underlying $Q$-value distribution due to the reward and $Q$-value jumps close to the teleporting tile.

Second, we consider a joint inventory replenishment problem (cf. Vanvuchelen et al., 2022). Consider a retailers' warehouse that stocks $N$ item types $i \in \mathcal{I}$. Each timestep, customer demand is served from the warehouse stock and the retailer needs to decide on the ordering quantity, aiming to minimize total costs that comprise per-item ordering costs $o_i$, holding costs $h_i$, and backorder costs $b_i$. The latter constitute a penalty for not being able to directly serve demand from stock in a time step. All individual items $i$ are linked together through a common order costs $O$. This fixed cost is incurred whenever at least one item is ordered, i.e., this term ensures that all items need to be considered simultaneously, since batch-reordering of multiple items is less costly. To ensure a finite decision space, we let the agent decide on *order-up-to levels*, which are bound by $S_{\max}$, i.e., $S_{\max}$ represents the maximum of items the retailer can stock. Hence, $|\mathcal{A}| = (S_{\max} + 1)^N$, which includes ordering zero items. For all our experiments we set $S_{\max}$ to 66, i.e., $|\mathcal{A}| = 67^N$.

Third, we consider a well-known dynamic job-shop scheduling problem (see, e.g., Zhang et al., 2020a; Wu & Yan, 2023). In this problem, a set of $G$ jobs is allocated to $N$ machines that can serve up to $L$ jobs each, yielding an action space of size $|\mathcal{A}| = L^N$. We consider a load-balancing variant of this problem, for which jobs need to be balanced over heterogeneous machines. Apart from a positive reward for each finished job, we incur negative rewards for an unbalanced allocation of jobs over machines and for energy consumption, which is heterogeneous over machines and is affected by the deterioration and repair of machines, as such adding stochasticity to the transition function. For our experiments, we set $G = 500$, $L = 100$, and $N = 5$.

## 5 NUMERICAL RESULTS

The following synthesizes the findings of our numerical studies, focusing on the scalability and algorithmic performance of each method. All results reported correspond to runs with the best

hyperparameters found for each method, executed over 50 seeds. We refer to the supplementary material for detailed results and information on the hyperparameter tuning. We compare the methods' step time and memory usage in Appendix H.

Table 1: Learning performance for different action space sizes $|\mathcal{A}|$.

|  | VAC | LAR | $k$nn | MinMax | DNC w/o SA | DNC |
|---|---|---|---|---|---|---|
| $10^3 < |\mathcal{A}| \leq 10^6$ | - | ✓ | ✓ | ✓ | ✓ | ✓ |
| $10^6 < |\mathcal{A}| \leq 10^9$ | - | - | ✓ | ◯ | ◯ | ✓ |
| $|\mathcal{A}| \gg 10^9$ | - | - | - | ◯ | ◯ | ✓ |

Table 1 compares algorithm performance in learning policies across different action space sizes in three environments. A checkmark denotes consistent learning of performant policies for all instances in a given action space size, while a circle indicates consistent learning for some instances, and a minus for none. VAC already struggles in spaces with thousands of actions, emphasizing the need for advanced methods to handle LDAS. For $10^6 < |\mathcal{A}| \leq 10^9$, $k$nn succeeds to learn a performant policy, but LAR fails to learn. LAR learns unique embeddings for each action, which is beneficial in unstructured action spaces where it identifies and uses a meta-structure. However, in SLDAS, creating embeddings for every action becomes intractable. For action spaces with $|\mathcal{A}| > 10^9$, $k$nn also fails due to memory constraints. Our DNC algorithm consistently learns performant policies across all sizes, showing robust scalability. The MinMax approach exposes difficulties in spaces larger than $10^6$, as it is susceptible to getting stuck in local optima as the action space grows.

The remainder of this section focuses on algorithmic performance, i.e., the average performance during testing. Here, we omit the analyses of algorithms that are not capable of learning performant policies at all for the respective environments. We report results for solving the MILP, which does not scale beyond the maze environment, in Appendix B. For all performant benchmarks, Figure 3 shows the expected test return evaluated after a different number of training iterations and averaged over 50 random seeds.

The two figures in the top row depict the results for the maze environment. We observe that VAC is unable to find a good policy already for "smaller" action spaces of size $2^{12}$ actions. The other algorithms do not differ significantly in their performance. We note that for this relatively

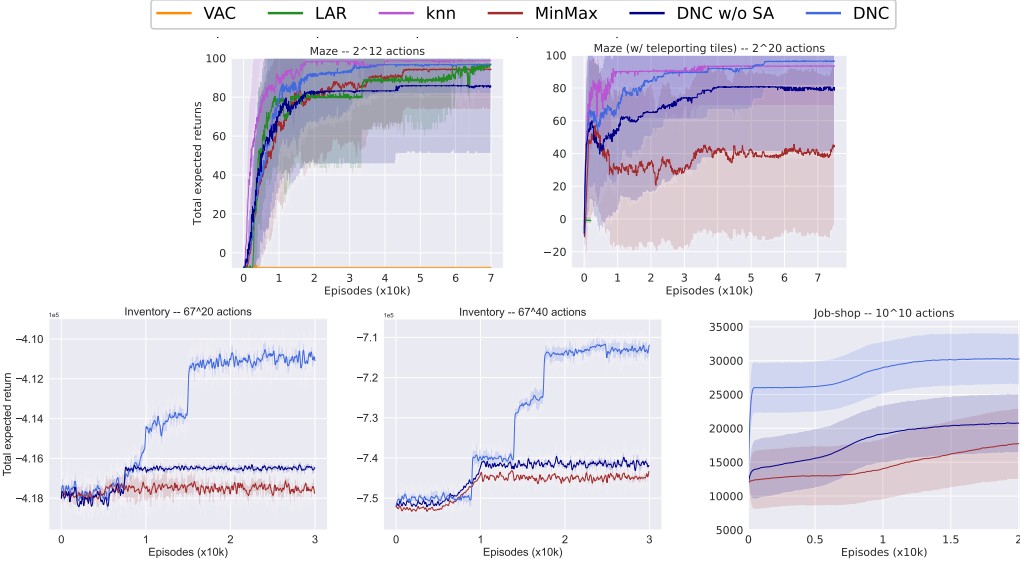

Figure 3: Average total expected returns during testing for 50 random seeds over the training iterations. The shaded area represents the 2 standard deviation training seed variance corridor. (Top) maze environment results. (Bottom) real-world environment results.

simple environment with low reward variance and smooth $Q$-value distribution, the benefit of considering neighbors to $\hat{a}$ is limited, such that DNC does not outperform MinMax but obtains a similar performance. Hence, if continuous actions can already be accurately mapped to discrete ones, evaluating neighbors has limited added value.

The results of the maze variant with a teleporting tile (second figure in the top row) show that methods incorporating neighborhood information (DNC, DNC w/o SA, $k$nn) perform better compared to MinMax. The latter exhibits unstable learning behavior, since the coarse rounding step cannot resolve sharp changes in $Q$-values caused by the teleporting tile. We conjecture that, when the underlying $Q$-value distribution of a problem is non-linear, the added benefit of considering more neighboring actions becomes more apparent. DNC is therefore able to provide high quality policies in more complex environments, comparable to $k$nn. This result is noteworthy, as MinMax is the only state-of-the-art method able to scale to very large SLDAS. We note that LAR did not complete the training required to learn action embeddings in the given time limit, hence only the first 2k episodes are shown for LAR. In the appendix, we provide boxplots showing the performance of the final learned policies. They suggest that convergence in the maze environment is much more stable than perceived in Figure 3, as the observed variance is the result of the maze's reward spread.

The bottom row of Figure 3 reports results on the inventory and job-shop environments. For the inventory environment, we observe that only DNC is able to learn a performant policy, while VAC, $k$nn and LAR simply run out of memory before finding a policy, and MinMax and DNC w/o SA converge to a considerably less performant policy. Similar to the maze environment with teleporting tiles, the joint inventory problem has a more non-linear $Q$-value function due to joint order costs, and neighboring actions can result in significantly different rewards. Hence, incorporating a neighborhood when selecting a discrete action and adding some randomness while exploring it proves advantageous.

The results for the job-shop scheduling environment (bottom-right graph in Figure 3) confirm the good performance of DNC. DNC outperforms its ablation DNC w/o SA and MinMax, while converging faster. This result demonstrates (i) the added value of considering multiple neighbors instead of only the nearest neighbor, highlighted by DNC w/o SA's advantage over MinMax, and (ii) the benefit of SA, highlighted by DNC's outperformance of DNC w/o SA. We observe that MinMax still exhibits a slight gradient at 20k episodes. To confirm that DNC outperforms MinMax after the latter's convergence, we present an additional visualization in Appendix H, Figure 11, providing evidence of MinMax's and DNC w/o SA's convergence at comparable performance after 80k episodes.

Concluding, our experimental results show that DNC matches or surpasses current benchmarks for all SLDAS experiments. It equals the top performance in simple scenarios and competes effectively with $k$nn and LAR – which use advanced learning representations – in complex cases with moderately large SLDAS featuring non-linear reward- and $Q$-value distributions. For very large SLDAS, DNC shows superior performance, since $k$nn and LAR cannot scale beyond enumerable action spaces, and the remaining benchmarks, MinMax and DNC w/o SA, are consistently outperformed.

## 6   CONCLUSION

In this paper, we present a novel algorithmic pipeline for deep reinforcement learning in environments with structured large discrete action spaces that enables to integrate an effective continuous-to-discrete action mapping in an actor critic algorithm. Our algorithmic pipeline shows two crucial advantages. First, it does not require enumerating the full action space, nor does it require storing the action space in-memory during the training process. Second, the perturbation scheme and search procedure allow for application to a wide range of problems that exhibit a structured action space. We compare our approach against various state-of-the-art benchmarks across three different environments. Our results show that our pipeline overcomes the benchmark's scalability limitations through a dynamic neighborhood evaluation. Our pipeline scales up to action space sizes up to $10^{73}$ and shows comparable or improving solution quality across the investigated environments.

Our algorithmic pipeline shows significant benefit over benchmarks: (i) when the discrete action space is very large yet structured; (ii) when the environment exhibits non-linearities in the reward- and $Q$-function. The promising results of DNC motivate future work to explore extensions of our pipeline to alternative neighborhood operators and selection criteria, efficient handling of infeasible actions and constraints, or using graph neural networks to efficiently evaluate neighborhoods.

## REPRODUCIBILITY STATEMENT

The code for running DNC and all benchmarks on the studied domains can be found in the attached supplement and at: `https://github.com/tumBAIS/dynamicNeighborhoodConstruction`. Proofs of all lemmata can be found in the supplement. Furthermore, the supplement details a comprehensive description considering the implementation of DNC in a standard actor-critic algorithm. Additionally, the supplement elaborates on the implementation of benchmarks.

## ACKNOWLEDGEMENTS

We would like to thank Yash Chandak for providing us with his implementation of the LAR approach. Moreover, we would like to thank Georgina Nouli for the discussions that helped us to substantiate the proofs for Lemmata 1, 2, and 3.

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

## A    Proofs of Lemmata 1, 2, and 3

**Lemma 1** *Action similarity $L$ is given by* $\displaystyle\sup_{\boldsymbol{a},\boldsymbol{a}'\in\mathcal{A}',\boldsymbol{a}\neq\boldsymbol{a}'}\frac{|Q^\pi(\boldsymbol{s},\boldsymbol{a})-Q^\pi(\boldsymbol{s},\boldsymbol{a}')|}{\|\boldsymbol{a}-\boldsymbol{a}'\|_2}$*, ensuring that* $|Q^\pi(\boldsymbol{s},\boldsymbol{a})-Q^\pi(\boldsymbol{s},\boldsymbol{a}')|\leq L\|\boldsymbol{a}-\boldsymbol{a}'\|_2$ *for all* $\boldsymbol{a},\boldsymbol{a}'\in\mathcal{A}'$.

**Proof**    Since the action neighborhood $\mathcal{A}'$ is finite and discrete, there exists a minimum Euclidean distance $\delta>0$ between any two distinct actions $\boldsymbol{a},\boldsymbol{a}'\in\mathcal{A}'$, i.e., $\|\boldsymbol{a}-\boldsymbol{a}'\|_2\geq\delta$ for $\boldsymbol{a}\neq\boldsymbol{a}'$.

Consider any two distinct actions $\boldsymbol{a},\boldsymbol{a}'\in\mathcal{A}'$ and their corresponding $Q$-values $Q^\pi(\boldsymbol{s},\boldsymbol{a})\in\mathbb{R}$ and $Q^\pi(\boldsymbol{s},\boldsymbol{a}')\in\mathbb{R}$, with state $\boldsymbol{s}$ fixed. By the triangle inequality, we have:

$$0\leq$$
$$|Q^\pi(\boldsymbol{s},\boldsymbol{a})-Q^\pi(\boldsymbol{s},\boldsymbol{a}')|=$$
$$|Q^\pi(\boldsymbol{s},\boldsymbol{a})+(-Q^\pi(\boldsymbol{s},\boldsymbol{a}'))|\leq$$
$$|Q^\pi(\boldsymbol{s},\boldsymbol{a})|+|-Q^\pi(\boldsymbol{s},\boldsymbol{a}')|=$$
$$|Q^\pi(\boldsymbol{s},\boldsymbol{a})|+|Q^\pi(\boldsymbol{s},\boldsymbol{a}')|\ .$$

Let $Q^{\max}=\max_{\boldsymbol{a}\in\mathcal{A}'}Q^\pi(\boldsymbol{s},\boldsymbol{a})$ be the maximum $Q$-value over all actions in $\mathcal{A}'$. From the triangle inequality, it follows that $|Q^\pi(\boldsymbol{s},\boldsymbol{a})-Q^\pi(\boldsymbol{s},\boldsymbol{a}')|\leq 2Q^{\max}$ must hold.

Now, for any action pair $\boldsymbol{a}\neq\boldsymbol{a}'$, we have $\|\boldsymbol{a}-\boldsymbol{a}'\|_2\geq\delta$ and $|Q^\pi(\boldsymbol{s},\boldsymbol{a})-Q^\pi(\boldsymbol{s},\boldsymbol{a}')|\leq 2Q^{\max}$. Hence, the ratio $\frac{|Q^\pi(\boldsymbol{s},\boldsymbol{a})-Q^\pi(\boldsymbol{s},\boldsymbol{a}')|}{\|\boldsymbol{a}-\boldsymbol{a}'\|_2}$ is a non-negative real number bounded by $\frac{2Q^{\max}}{\delta}$ for all $\boldsymbol{a},\boldsymbol{a}'\in\mathcal{A}'$ satisfying $\boldsymbol{a}\neq\boldsymbol{a}'$. Since the action space is finite, the number of ratios is also finite, and we bound the Lipschitz constant:

$$L=\sup_{\boldsymbol{a},\boldsymbol{a}'\in\mathcal{A}',\boldsymbol{a}\neq\boldsymbol{a}'}\frac{|Q^\pi(\boldsymbol{s},\boldsymbol{a})-Q^\pi(\boldsymbol{s},\boldsymbol{a}')|}{\|\boldsymbol{a}-\boldsymbol{a}'\|_2}\leq\frac{2Q^{\max}}{\delta}\ .$$

Hence, $L$ exists and is finite, providing a measure on action similarity.    $\square$

**Lemma 2** *If $J$ is locally upward convex for neighborhood $\mathcal{A}'$ with maximum perturbation $(d\,\epsilon)$ around base action $\bar{a}$, then worst-case performance with respect to $\bar{a}$ is bound by the maximally perturbed actions $a'' \in \mathcal{A}''$ via $Q^\pi(s, a') \geq \min\limits_{a'' \in \mathcal{A}''} Q^\pi(s, a''), \forall a' \in \mathcal{A}'.$*

Figure 4 illustrates the intuition behind local convexity and our proof.

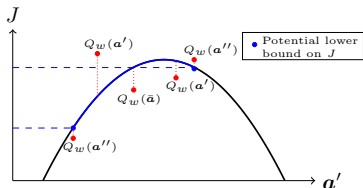

Figure 4: Illustration of a locally convex neighborhood of $J$ with respect to the perturbed actions $a'$. DNC selects actions based on $Q(a')$ and thus may return any $a' \in \mathcal{A}'$. The convex property guarantees that $J(a'') \leq J(a'), \forall a' \in \mathcal{A}'$ for some maximally perturbed action $a''$.

**Proof**   Evaluating $\arg\max\limits_{a \in \mathcal{A}} Q_w(s, a)$ may return any $a \in \mathcal{A}'$, as $Q_w$ might have arbitrary values. Therefore, we must prove that the performance bound holds for all $a \in \mathcal{A}'$. Via the local convexity of $J$ around base action $\bar{a}$, we prove that worst-case performance is bound by a maximally perturbed action in $\mathcal{A}''$.

Let $\lambda \in [0, 1]$, $a'', a''', a'''' \in \mathcal{A}''$ and $a' \in \mathcal{A}'$. By definition of upward convexity, the following inequalities are satisfied.

$$\min_{a'' \in \mathcal{A}''} Q^\pi(s, a'') =$$
$$\lambda \min_{a'' \in \mathcal{A}''} Q^\pi(s, a'') + (1 - \lambda) \min_{a'' \in \mathcal{A}''} Q^\pi(s, a'') \leq$$
$$\lambda Q^\pi(s, a''') + (1 - \lambda) Q^\pi(s, a'''') \leq$$
$$Q^\pi(s, \lambda(a''') + (1 - \lambda)(a'''')) .$$

This result holds $\forall \lambda \in [0, 1]$ and all maximally perturbed actions. Now, we only need to prove that $\forall a' \in \mathcal{A}' \exists (\lambda, a''', a'''')$ such that $a' = \lambda a''' + (1 - \lambda) a''''$, i.e., that linear combination of maximally perturbed actions can express all feasible actions $a' \in \mathcal{A}'$.

Let us express neighbors via $a' = \bar{a} + P_j$ for some $j \in \{1, \dots, 2Nd\}$. Moreover, let $j^+ \in \{Nd - (N - 1), \dots, Nd\}$ be column indices corresponding to maximally positively perturbed actions and let $j^- \in \{2Nd - (N - 1), \dots, 2Nd\}$ correspond to maximally negatively perturbed actions, and let $l$ be the index of the non-zero entry of $P_j$. Then, consider the maximally perturbed actions $A^l_{\cdot j^+}, A^l_{\cdot j^-}$, who only differ from $\bar{a}$ on their $l$th element. We can now express $a'$ as

$$a' = \lambda A^l_{\cdot j^+} + (1 - \lambda) A^l_{\cdot j^-}. \tag{5}$$

Here, we obtain the required $\lambda$ by solving Equation equation 5 for $\lambda$, for which we use the relevant perturbed entry $P_{lj}$. Solving the equation leads to $\lambda = \frac{P_{lj} + d\,\epsilon}{2\,d\,\epsilon}$, yielding a value between 0 and 1 as $-(d\,\epsilon) \leq P_{lj} \leq (d\,\epsilon)$. Therefore, we can express all neighbors $a' \in \mathcal{A}'$ as linear combinations of maximally perturbed actions; hence, $\min\limits_{a'' \in \mathcal{A}''} Q^\pi(s, a'') \leq Q^\pi(s, a'), \forall a' \in \mathcal{A}'$.   $\square$

**Lemma 3** *Consider a neighborhood $\mathcal{A}'$ and improving actions satisfying $Q^\pi(s, a) > \max\limits_{a' \in \mathcal{A}'} Q^\pi(s, a'), a \in \mathcal{A} \setminus \mathcal{A}'$. In finite time, DNC will accept improving actions, provided that (i) $\beta$ and $k$ cool sufficiently slowly and (ii) a maximum perturbation distance $(d\,\epsilon)$ is set such that all action pairs can communicate.*

**Proof** The proof is structured into three steps: first, we show that DNC's simulated annealing procedure behaves as a non-homogeneous Markov chain that searches over the action space. We then show that this Markov chain is aperiodic and irreducible. Finally, we show that an arbitrary action can be reached with positive probability in a finite number of steps.

*1. Simulated annealing procedure behaves as a Markov chain*
We first establish the preliminaries of the simulated annealing procedure used in DNC, which is necessary to describe the procedure as a Markov chain (cf. Bertsimas & Tsitsiklis, 1993).

1. $\mathcal{A}$ is a finite discrete action set.

2. There exists a real-valued cost function $Q^\pi : \mathcal{S} \times \mathcal{A} \mapsto \mathbb{R}$, with a proper subset of local minima $\mathcal{A}^* \subset \mathcal{A}$. Specifically, we associate each state-action pair with an estimated $Q$-value $Q_w(\boldsymbol{s}, \boldsymbol{a}) \in \mathbb{R}$. Since state $\boldsymbol{s}$ is fixed while executing the simulated annealing algorithm, we omit its notation moving forward.

3. Every action $\boldsymbol{a} \in \mathcal{A}$ has a non-empty neighborhood $\mathcal{A}_{\boldsymbol{a}} \subseteq \mathcal{A}$ that includes itself, such that $|\mathcal{A}_{\boldsymbol{a}}| > 1$. This can be ensured by setting an appropriate maximum perturbation distance $(d\,\epsilon) \in \mathbb{R}^+$. As DNC performs perturbations on each individual entry in the action vector, it follows that all feasible entries (and thus all actions) in the finite action space can be constructed through perturbation of neighboring entries. Finally, given that the Euclidean distance is a symmetric metric, it follows that $\boldsymbol{a}' \in \mathcal{A}_{\boldsymbol{a}} \iff \boldsymbol{a} \in \mathcal{A}_{\boldsymbol{a}'}$.

4. When we find non-improving neighbors $\boldsymbol{k}_1$ – which happens in finite time given that the action space is finite – there exists a set of positive probabilities $p_{\boldsymbol{a},\boldsymbol{a}'}, \boldsymbol{a} \neq \boldsymbol{a}'$ that reflect the probability of evaluating neighbor $\boldsymbol{a}'$ from $\boldsymbol{a}$. The sum of probabilities satisfies $\sum_{\boldsymbol{a}' \in \mathcal{A} \setminus \{\boldsymbol{a}\}} p_{\boldsymbol{a},\boldsymbol{a}'} = 1$. Specifically, in (l.14) of Algorithm 1, we associate $\frac{1}{|\mathcal{K}|}$ to each $k_{\mathrm{rand}} \in \mathcal{K}$. As mentioned, we always reach (l.14) in finite time, due to the guarantee of finding non-improving actions in a finite space.

5. There is a temperature scheme $\beta : \mathbb{N} \mapsto (0, \infty)$, with $\beta_t$ representing temperature at time $t \in \mathbb{N}$ and $\beta_t \geq \beta_{t+1}, \forall t$. Similarly, the scheme $k : \mathbb{N} \mapsto [0, |\mathcal{A}|]$ returns the number of neighbors $k_t$ generated at time $t$, with $k_t \geq k_{t+1}, \forall t$. As a preliminary for the remainder of the proof, the temperature must cool sufficiently slow to allow finite-time transitions between any $\boldsymbol{a}, \boldsymbol{a}' \in \mathcal{A}$.

6. At $t = 0$, an initial action $\bar{\boldsymbol{a}}$ is given. This action is generated by the continuous-to-discrete mapping function $g : \hat{\boldsymbol{a}} \mapsto \bar{\boldsymbol{a}}$, as detailed in the paper.

Given these preliminaries, we define the simulated annealing procedure used in our DNC as a non-homogeneous Markov chain $A = A_0, A_1, A_2, \ldots$ that searches over $\mathcal{A}$. Here, $A_t$ is a random variable that denotes the accepted action after move $t$, i.e., $A_t = \boldsymbol{a}_t, \boldsymbol{a}_t \in \mathcal{A}$.

Let us denote $k_1 = \underset{k \in \mathcal{K}'}{\arg\max} \left( Q(s, k) \right)$ and $\kappa = \max\left( 0, \frac{Q(s,a) - Q(s,k_1)}{\beta} \right)$. Then, from the algorithmic outline, we derive that the probability of $A_{t+1} = \boldsymbol{a}'$ when $A_t = \boldsymbol{a}$ is given by

$$\mathbb{P}(A_{t+1} = \boldsymbol{a}' | A_t = \boldsymbol{a}) = \begin{cases} \exp(\kappa) + (1 - \exp(\kappa)) \cdot \frac{1}{|\mathcal{K}|}, & \text{if}, \boldsymbol{a}' = \boldsymbol{k}_1 \\ (1 - \exp(\kappa)) \cdot \frac{1}{|\mathcal{K}|}, & \text{if}, \boldsymbol{a}' \neq \boldsymbol{k}_1 \end{cases} \tag{6}$$

*2. Markov chain is irreducible and aperiodic*
We now show that the transition probabilities of the Markov chain imply it is (i) irreducible and (ii) aperiodic, which are necessary and sufficient conditions to prove that $\mathbb{P}(A_{t+\tau} = \boldsymbol{a}' | A_t = \boldsymbol{a}) > 0$ for some finite $\tau \in \mathbb{N}$.

(i) *Irreducibility of $A$*: To show that the Markov chain is irreducible, suppose we set the maximum perturbance distance to $(d\,\epsilon) = \underset{\boldsymbol{a}, \boldsymbol{a}' \in \mathcal{A}}{\max} \|\boldsymbol{a} - \boldsymbol{a}'\|_2$, i.e., equaling the finite upper bound on perturbation. Now, suppose we wish to move between arbitrary actions $\boldsymbol{a} = (a_n)_{\forall n \in \{1, \ldots, N\}}$ and $\boldsymbol{a}' = (a'_n)_{\forall n \in \{1, \ldots, N\}}$. The chosen perturbation distance ensures that action entries can be perturbed to any target value $a'_n \in \{a_n - (d\,\epsilon), \ldots, a_n + (d\,\epsilon)\}$. By perturbing each entry $a_n$ individually, we can reach $\boldsymbol{a}'$ within $N$ steps, as Equation equation 6 ensures a positive probability of accepting such perturbations. To generalize the established

result, observe that we may relax to $(d\,\epsilon) \leq \max\limits_{\boldsymbol{a}, \boldsymbol{a}' \in \mathcal{A}} \|\boldsymbol{a} - \boldsymbol{a}'\|_2$ and can construct a similar rationale for some smaller $d$ that satisfies communication between action pairs as well.

(ii) *Aperiodicity of $A$*: As we accept non-improving actions with a probability $< 1$ and $\exists \boldsymbol{a}^* \in \mathcal{A}^*$ with no improving neighbors, a one-step transition probability $\mathbb{P}(A_{t+1} = \boldsymbol{a}^* | A_t = \boldsymbol{a}^*) > 0$ is implied by Equation equation 6. By definition of aperiodicity, identifying one aperiodic action suffices to prove that the entire Markov chain is aperiodic.

*3. All actions are reachable with positive probability in finite time*
We have shown that the Markov chain is irreducible and aperiodic, proving that all actions belong to the same communicating class. By the Perron-Frobenius theorem, $\forall(\boldsymbol{a}, \boldsymbol{a}')$, there exists a finite $\tau$ such that $\mathbb{P}(A_{t+\tau} = \boldsymbol{a}' | A_t = \boldsymbol{a}) = (\mathcal{P}^\tau)_{\boldsymbol{a}\boldsymbol{a}'} > 0$, where $(\mathcal{P}^\tau)_{\boldsymbol{a}\boldsymbol{a}'}$ is a $\tau$-step transition matrix.

This result shows that there is a positive probability of reaching action $\boldsymbol{a}'$ from action $\boldsymbol{a}$ in $\tau$ steps. Thus, given sufficiently large perturbation distances and an appropriate cooling scheme, DNC enables to find improving actions outside the initial neighborhood. □

## B  FULL SET OF MILP CONSTRAINTS AND FURTHER RESULTS

In addition to Constraints 2 to 4, the following constraints are required to describe the DNN ($\forall l \geq 3$)

$$y_{d_l} \geq \sum_{d_{l-1} \in \mathcal{D}_{l-1}} w_{d_{l-1}, d_l}\, y_{d_{l-1}} \tag{7}$$

$$y_{d_l} \leq z_{d_l}\, M \tag{8}$$

$$y_{d_l} \leq (1 - z_{d_l})\, M + \sum_{d_{l-1} \in \mathcal{D}_{l-1}} w_{d_{l-1}, d_l}\, y_{d_{l-1}} \tag{9}$$

$$z_{d_l} \geq \frac{\sum_{d_{l-1} \in \mathcal{D}_{l-1}} w_{d_{l-1}, d_l}\, y_{d_{l-1}}}{M} \tag{10}$$

$$z_{d_l} \leq 1 + \frac{\sum_{d_{l-1} \in \mathcal{D}_{l-1}} w_{d_{l-1}, d_l}\, y_{d_{l-1}}}{M} \tag{11}$$

$$z_{d_l} \in \{0, 1\}. \tag{12}$$

These technical constraints are required to describe and ensure consistency of the ReLU values of the DNN, as described in Bunel et al. (2017) or van Heeswijk & La Poutré (2020). Moreover, to further bound and efficiently solve the MILP, we introduce the following branching constraints, which bounds the maximum Hamming distance $k$ between the base action $\bar{\boldsymbol{a}}$ and the resulting optimal action $\bar{\boldsymbol{a}}^*$, hence, further restricting the neighborhood and consequently the search space (Fischetti & Lodi, 2003):

$$k \geq \sum_{j:\bar{a}_j = a_j^{l'}} \mu_j \left( \bar{a}^* - a_j^{l'} \right) + \sum_{j:\bar{a}_j = a_j^{u'}} \mu_j \left( a_j^{u'} - \bar{a}^* \right) + \sum_{j:a_j^{k'} < \bar{a}_j < a_j^{u'}} \mu_j \left( a_j^+ + a_j^- \right), \tag{13}$$

where $\mu_j = \frac{1}{a_j^{u'} - a_j^{l'}}$ and

$$\bar{a}_j^* = \bar{a}_j + a_j^+ - a_j^-; \qquad a_j^+ \geq 0,\, a_j^- \geq 0; \qquad \forall j : a_j^{l'} < \bar{a}_j^* < a_j^{u'}. \tag{14}$$

We report results for two approaches involving solving the MILP. We solve the MILP using Gurobi 10.00 and report results on the maze environment with an action space size of $2^{12}$ in Figure 5. Figure 5a shows the results for solving the MILP during both training and testing (pure MILP) as well as the results of using DNC during training and solving the MILP during testing only. We refer to the latter approach as *hybrid*. First, we observe that if we solve the MILP during both training and testing, the time limit is reached before convergence (cf. Figure 5a). When using the hybrid approach, we observe a worse performance compared to DNC. This performance difference between DNC and hybrid may be explained by the fact that DNC is able to find more distant actions in terms of Hamming and/or Euclidean distance by means of the SA procedure, whereas the MILP is confined to the original neighborhood's bounds. The results indicate that DNC benefits from the SA procedure's ability to escape the initial (user-defined) perturbation boundaries in case the best neighbor happens to be outside them.

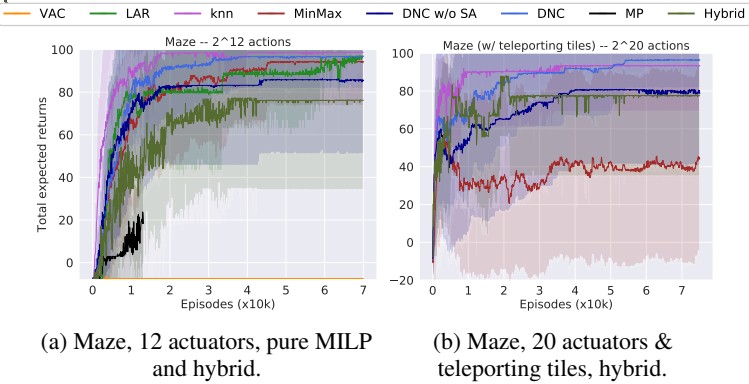

(a) Maze, 12 actuators, pure MILP and hybrid.

(b) Maze, 20 actuators & teleporting tiles, hybrid.

Figure 5: Average total expected returns during testing for 50 random seeds over the training iterations in the maze environment, including the performance of MILP and a hybrid approach. The shaded area represents the 2 standard deviation training seed variance corridor.

## C    ENVIRONMENTS

**Maze**    Our implementation follows the implementation as described in Chandak et al. (2019a). We set the episode length to $150$ steps and provide a reward of $-0.05$ for each move and $100$ for reaching the target. The target, wall, and initial agent position follow the illustration in Figure 6a. Here, the red dot represents the agent with the actuators, the grey areas are walls which the agent cannot move through, and the yellow star is the target area. The agent cannot move outside the boundaries of the maze. Whenever a wall or outside boundary is hit, the agent does not move and remains in the same state. Action noise was added to make the problem more challenging. On average $10\%$ of the agent movements are distorted by a noise signal, making the agents' movements result in slightly displaced locations. In Figure 6b we report the variation of the maze environment, wherein the agent starts in the lower right corner, receives a reward of +100 when reaching the target, a reward of -0.05 for each step, and a reward of -20 when entering the teleporting tile (grey rectangle in the middle). When entering the teleporting tile the agent's position is reset to the lower right corner. This environment has the same action noise as in the original maze environment.

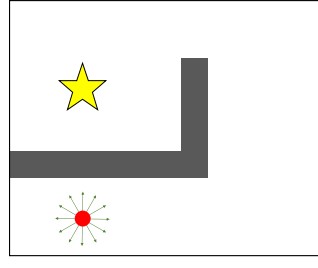

(a) Maze with wall.

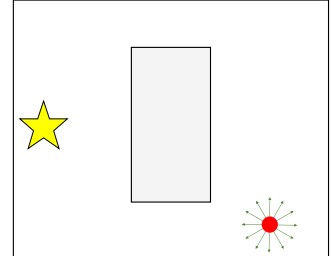

(b) Maze with teleporting tile in the middle.

Figure 6: Illustration of the maze environment.

**Inventory Replenishment**    We mostly follow the implementation as detailed in Vanvuchelen et al. (2022), wherein they consider a retailer managing uncertain demand for different items in its warehouse. Compared to them, we run shorter episodes to decrease the computational burden. The costs per item $i \in \mathcal{I}$ are as follows: holding costs $h_i = 1$, backorder costs $b_i = 19$, ordering costs $o_i = 10$ and the common order costs are $O = 75$. The order-up-to levels are set to the range $[0, 66]$. We sample the demand rate from the Poisson distribution, with half of the items having a demand rate of $\lambda_i = 10$, and the other half of the items $\lambda_i = 20$. We initialize all inventory levels to 25. Every

episode comprises 100 timesteps. The reward function can be denoted by:

$$R_t = \sum_{i=1}^{N} \left( h_i I_{i,t}^+ + b_i I_{i,t}^- + o_i a_i \right) + O\mathbb{1}_{\{\sum_{i=1}^{N} q_{i,t} > 0\}} \ , \tag{15}$$

where $I_{i,t}^+$ and $I_{i,t}^-$ indicate all positive and negative stock levels, respectively. $\mathbb{1}_{\{\sum_{i=1}^{N} q_{i,t} > 0\}}$ is the indicator function that indicates if at least one product is ordered.

**Job-shop scheduling**   Each machine $i$ has a different energy consumption per job. Whenever the number of jobs on a machine is higher than $75\%$ or lower than $50\%$ of $L$ (the maximum number of jobs per machine) the machine's deterioration factor $w_i$ is increased or reduced, respectively. The deterioration factor directly impacts the efficiency of machines, and hence, it affects the energy consumption per job. We consider the increase in deterioration to be caused by wear, i.e., over-use, and the reduction of the deterioration to be achieved through machine repair, which are only possible when the utilization of the machine is low. Deterioration and repairs are stochastic and drawn from a uniform distribution. Deterioration causes an increase of the deterioration factor, bounded by $\Delta w_i \in [0.05, 0.4]$. Repair causes a decrease in deterioration factor, $\Delta w_i \in [-0.4, -0.05]$. The deterioration factor $w_i$ is multiplied by the energy consumption $b_i$ to obtain the energy consumption per job. The deterioration factor $w_i$ of all machines is 1.0 in the first time step, and can never drop below 1.0 through repairs. We bound the energy consumption per machine by $M$. The reward function is given by:

$$R_t = \sum_{i=1}^{N} \left( a_i p - \min\{a_i b_i w_i, M\} \right) - \sigma(a), \tag{16}$$

where $a_i$ are the number of jobs allocated to machine $i$, $p$ is the reward for each finished job, and $\sigma(a)$ is the load-balancing reward that provides a negative reward for an unbalanced allocation using the standard deviation of the allocated jobs to machines. We set the reward factors to $p = 3$ and $M = 100$. The value of $L$ is 100. We sample the energy consumption $b_i$ per machine from the uniform distribution, with bounds $[1.0, 2.0]$. The results in the paper correspond to a problem with $N = 5$ machines.

## D   Implementation Details

**Integration of DNC in an actor-critic algorithm**   Algorithm 2 details the integration of DNC into an actor-critic reinforcement learning (RL) algorithm. Specifically, we initialize the network weights $\boldsymbol{w}$ and $\boldsymbol{\theta}$ of the critic and actor respectively (l.1), and set hyperparameters such as the Gaussian $\boldsymbol{\sigma}$ and the critic- and actor learning rates $\alpha_{\mathrm{cr}}$ and $\alpha_{\mathrm{ac}}$ (l.2). After initializing a state $\boldsymbol{s}$ (l.4), we loop through each time step of an episode (l.5). We obtain a continuous action $\hat{\boldsymbol{a}}$ by sampling it from $\pi_{\boldsymbol{\theta}}$ according to the learned $\boldsymbol{\mu}_{\boldsymbol{\theta}}$ and $\boldsymbol{\sigma}$ of a Gaussian distribution and the hyperparameter $\boldsymbol{\theta}$ (l.6). Next, we obtain a discrete action $\bar{\boldsymbol{a}}^*$ by applying DNC (l.7), whose details are provided in Algorithm 1 in the main body of the paper. We then apply $\bar{\boldsymbol{a}}^*$ to the environment, observe reward $r$ and next state $\boldsymbol{s}'$ (l.8). We obtain the next state's continuous action $\hat{\boldsymbol{a}}'$ (l.9) and then its discrete action $\bar{\boldsymbol{a}}^{*'}$ (l.10). Using losses based on the observed TD-error (l.11), we update critic (l.12) and actor (l.13) weights. Note that we use both $\hat{\boldsymbol{a}}$ and a TD-error based on $\bar{\boldsymbol{a}}^*$ and $\bar{\boldsymbol{a}}^{*'}$, hence using slightly off-policy information to compute the actor loss. However, since in practice DNC does not move far away from $\hat{\boldsymbol{a}}$, using off-policy information in the actor weight update does not heavily impact on learning stability.

**Details on the Neural Network Architecture**   When applying a deep network architecture, we use two hidden layers with ReLU activation functions for both actor and critic for DNC and all benchmarks. In Section F we provide further details for the architecture per environment. For all environments, we use a tanh output layer for the actor and do not bound the output of the critic.

We train the actor and critic based on the stochastic gradient descent algorithm implemented in PyTorch and use a Huber loss to train the critic and ensure stable weight updates.

**Other Implementation Details**   We employ a discretization function $g(\hat{\boldsymbol{a}})$ to obtain a discrete base action $\bar{\boldsymbol{a}}$ from the continuous action $\hat{\boldsymbol{a}}$.

---

**Algorithm 2** Actor critic pseudo-code with DNC.

---

1: Initialize network weights $\boldsymbol{w}, \boldsymbol{\theta}$
2: Set hyperparameters: $\boldsymbol{\sigma}, \alpha_{\mathrm{cr}}, \alpha_{\mathrm{ac}}$
3: **for** each episode **do**
4:     Initialize $\boldsymbol{s}$
5:     **for** each time step $t$ **do**
6:         $\hat{\boldsymbol{a}} \leftarrow \pi_{\boldsymbol{\theta}}(\boldsymbol{s})$ (based on $\boldsymbol{\sigma}$)
7:         $\bar{\boldsymbol{a}}^* = \mathrm{DNC}(\hat{\boldsymbol{a}})$ (see Algorithm 1)
8:         Apply $\bar{\boldsymbol{a}}^*$ to environment, observe reward $r$, and successor state $\boldsymbol{s}'$
9:         $\hat{\boldsymbol{a}}' \leftarrow \pi_{\boldsymbol{\theta}}(\boldsymbol{s}')$ (based on $\boldsymbol{\sigma}$)
10:        $\bar{\boldsymbol{a}}^{*'} = \mathrm{DNC}(\hat{\boldsymbol{a}}')$
11:        $\delta = r + \gamma\, Q(\boldsymbol{s}', \bar{\boldsymbol{a}}^{*'}, \boldsymbol{w}) - Q(\boldsymbol{s}, \bar{\boldsymbol{a}}^*, \boldsymbol{w})$
12:        $\boldsymbol{w} \leftarrow \boldsymbol{w} - \alpha_{\mathrm{cr}} \nabla_{\boldsymbol{w}} \delta$
13:        $\boldsymbol{\theta} \leftarrow \boldsymbol{\theta} + \alpha_{\mathrm{ac}} \delta \nabla_{\boldsymbol{\theta}} \log \pi_{\boldsymbol{\theta}}(\boldsymbol{s}, \hat{\boldsymbol{a}})$

---

$$g(\hat{a}) = \left\lfloor \frac{\mathrm{clip}(\hat{a}) - c_{\min}}{c_{\max} - c_{\min}} \cdot (a_{\max} - a_{\min}) + a_{\min} \right\rceil$$

with

$$\mathrm{clip}(\hat{a}) = \begin{cases} c_{\min}, & \text{if } \hat{a} < c_{\min}, \\ c_{\max}, & \text{if } \hat{a} > c_{\max}, \\ \hat{a}, & \text{otherwise,} \end{cases}$$

denoting a clipping function, as similar to Vanvuchelen et al. (2022). The discretization function has multiple hyperparameters that need to be selected based on (i) the output layer activation function, and (ii) the action dimension. These hyperparameters are $c_{\min}, c_{\max}, a_{\min}$, and $a_{\max}$. Since the action is sampled from a distribution, an action might overflow and result in non-existent actions, hence, clipping is required. Each value needs to be set to an appropriate value. We use $c_{\min} = -1, c_{\max} = 1$ for clipping, since we employ a tanh activation function for the output layer. The values for $a_{\min}$ and $a_{\max}$ are set depending on the environment; we use $[0, 1]$ for maze, $[0, 66]$ for the inventory environment, and $[0, 100]$ for the job-shop environment.

We represent states across all environments by means of a Fourier basis as described in Konidaris et al. (2011) and applied in Chandak et al. (2019a). In the maze environment we use a Fourier basis of order three with coupled terms. For the inventory and job-shop environments, we use decoupled terms to maintain a reasonable state vector size.

## E  BENCHMARKS

We consider four benchmarks and an ablation: VAC, MinMax (MinMax), $k$nn, LAR, and DNC without SA. For a full and detailed explanation of the benchmarks, we refer to Sutton & Barto (2018), Vanvuchelen et al. (2022), Dulac-Arnold et al. (2015), and Chandak et al. (2019a), respectively. Here, we restrict ourselves to a short description of each benchmark and explain how we embed these benchmarks in an actor-critic algorithm similar to Algorithm 2.

**VAC**    We employ a standard actor-critic method as baseline. For this method we employ a categorical policy, i.e., $\pi$ describes the probability of taking action $\boldsymbol{a}$ when being in state $\boldsymbol{s}$. We denote the policy by $\pi(\boldsymbol{a}|\boldsymbol{s})$ to emphasize that $\pi$ is a distribution. VAC's implementation follows Algorithm 2, as detailed above, with the only difference that we obtain the discrete action directly from the actor, instead of obtaining a continuous action and subsequently using DNC.

**MinMax**    The MinMax benchmark uses an actor-critic framework, in which the actor outputs a continuous action vector and function $g$ is applied in the same way as for DNC. We obtain the algorithm corresponding to MinMax by exchanging DNC in lines 7 and 10 of Algorithm 2 by $g$.

$k$**nn**    The $k$nn approach uses an approximate nearest neighbor lookup (Muja & Lowe, 2014) to find discrete neighbors in the space $\mathcal{A}$ based on continuous action $\hat{\boldsymbol{a}}$. The mapping function $h$ finds the $k$

nearest discrete neighbors in terms of Euclidean distance:

$$h_k(\hat{\boldsymbol{a}}) = \arg\min_{\boldsymbol{a} \in \mathcal{A}^k} \|\boldsymbol{a} - \hat{\boldsymbol{a}}\|_2.$$

After finding $k$ neighbors, the neighbor with highest $Q$-value is chosen and applied to the environment, using a similar approximate on-policy rationale as applied for DNC. Note that the critic is only used to select an action *after* all neighbors have been generated. In Section F we present different values of $k$, over which we search for the best performing hyperparameter setting for $k$nn. To embed $k$nn in an actor-critic algorithm, we modify Algorithm 2 in lines 7 and 10 as follows: instead of applying DNC, we search for the $k$-nearest neighbors of $\hat{\boldsymbol{a}}$ and $\hat{\boldsymbol{a}}'$, and, subsequently, obtain $\bar{\boldsymbol{a}}^*$ and $\bar{\boldsymbol{a}}^{*'}$ by selecting the neighbor with highest $Q$-value.

**LAR**  To setup the LAR benchmark, we use the code that implements the work presented in Chandak et al. (2019a) and that was kindly shared with us. In the following we briefly describe the algorithm. Before training the RL agent, we apply an initial supervised learning process to learn unique action embeddings $\boldsymbol{e}' \in \mathbb{R}^l$ for each discrete action $\boldsymbol{a}$. We use a buffer that stores state, action, and successor state transition data to feed the supervised learning model. We set the maximum buffer size to $6e5$ transitions and obtain these transitions from, e.g., a random policy. The supervised loss is determined using the KL-divergence between the true distribution $\mathbb{P}(\boldsymbol{a}_t|\boldsymbol{s}_t, \boldsymbol{s}_{t+1})$ and the estimated distribution $\hat{\mathbb{P}}(\boldsymbol{a}_t|\boldsymbol{s}_t, \boldsymbol{s}_{t+1})$, which describe probabilities of taking an action $\boldsymbol{a}_t$ at time step $t$ when having a certain state tuple $(\boldsymbol{s}_t, \boldsymbol{s}_{t+1})$. Across all environments, we use a maximum of 3000 epochs to minimize the supervised loss. We note here that the training process always converged before reaching the 3000 epochs limit. We detail the different sizes of the two-layer neural network architecture, used in the supervised learning procedure, in Section F. Moreover, note that the size of $\boldsymbol{e}$ may be both larger or smaller than the discrete action's size. It is up to the user to determine the embedding size, hence, it is a hyperparameter whose values we also report in Section F.

Following the initial supervised learning process, we proceed in a similar manner to Algorithm 2. First, we obtain $\boldsymbol{e}$ from the continuous policy $\pi$ (cf. line 6 in Algorithm 2). Second, we find the embedding $\boldsymbol{e}'$ closest to $\boldsymbol{e}$ based on an $L_2$ distance metric and look up the discrete action $\boldsymbol{a}$ corresponding to $\boldsymbol{e}'$ (cf. line 7 in Algorithm 2). At the end of each step of the episode, we update the continuous representations $\boldsymbol{e}'$ of $\boldsymbol{a}$ by performing one supervised learning step.

**DNC without SA**  Algorithm 3 details the DNC algorithm without the SA procedure, as used for our ablation study. First, we obtain a discrete base action $\bar{\boldsymbol{a}}$ (l.1) and find its discrete neighbors by DNC (l.2). We obtain the critic's $Q$-values (l.3), and subsequently select the action with the highest $Q$-value (l.4).

---

**Algorithm 3** Dynamic Neighborhood Construction w/o SA

---

1: Initialize $\bar{\boldsymbol{a}} \leftarrow g(\hat{\boldsymbol{a}}), \bar{\boldsymbol{a}}^* \leftarrow \bar{\boldsymbol{a}},$
2: Find neighbors $\mathcal{A}'$ to $\bar{\boldsymbol{a}}$ with $P_{ij}$
3: Get $Q$-values for all neighbors in $\mathcal{A}'$
4: $\bar{\boldsymbol{a}}^* \leftarrow \mathcal{A}'$, with $\bar{\boldsymbol{a}}^* = \arg\max_{a \in \mathcal{A}'} Q_w(\boldsymbol{s}, \boldsymbol{a})$
5: Return $\bar{\boldsymbol{a}}^*$

---

# F  HYPERPARAMETERS

In this section, we detail the hyperparameter settings used across the different environments. To this end, we provide an overview of hyperparameter settings in Table 2 and discuss specific settings that we use over all environments. N/A indicates that the respective method did not yield a performant policy for any hyperparameter setting.

We applied a search over the reported set of values in Table 2 (column "Set of values") to choose the best hyperparameter setting. Note that a value of zero nodes of the critic and actor layer corresponds to a shallow network. Moreover, we do not report hyperparameter settings for MinMax separately, as its only hyperparameter corresponds to $c_{\min}$ and $c_{\max}$, which we set to the minimum, resp. maximum value of the actor's output layer as described in Section D.

We study both (i) learning the second moment of the Gaussian distribution, $\sigma$, and (ii) setting $\sigma$ to a constant value. In Table 2, a † indicates that $\sigma$ was learned by the actor. We found that a constant $\sigma$ often led to faster convergence without performance loss.

The DNC-specific parameters are (i) the neighborhood depth $d$, (ii) the $k$-best neighbors to consider, (iii) the acceptance probability parameter $\beta$, and (iv) the cooling parameter $c$. We do not tune $\beta$, and set it to an initial value of $0.99$. The cooling parameter expresses by how much the parameter $k$ and $\beta$ are decreased every iteration of the search, in terms of percentage of the initial values of both parameters. We set $k$ respective to the size of neighborhood $|\mathcal{A}'|$.

Table 2: Hyperparameters, set of values, and chosen values.

| | Hyperparameters | Set of values | Chosen values | | | |
| | | | Maze | Maze w/ teleport tiles | Inventory | Job-shop |
|---|---|---|---|---|---|---|
| Overall | $\alpha_{cr}$ (critic learning rate) | $\{10^{-2}, 10^{-3}, 10^{-4}\}$ | $10^{-2}$ | $10^{-2}$ | $10^{-3}$ | $10^{-4}$ |
| | $\alpha_{ac}$ (actor learning rate) | $\{10^{-3}, 10^{-4}, 10^{-5}\}$ | $10^{-2}$ | $10^{-2}$ | $10^{-4}$ | $5 \cdot 10^{-5}$ |
| | $\sigma$ | $\{\dagger, 0.25, 0.5, 1\}$ | 1 | 1 | 1 | 1 |
| | # actor NN nodes/layer | $\{0, 32, 64, 128\}$ | 0 | 0 | 64 | 0 |
| | # critic NN nodes/layer | $\{0, 32, 64, 128\}$ | 32 | 32 | 128 | 32 |
| LAR | $|e|$ | $\{0.5\,|s|, |s|, 2\,|s|\}$ | $|s|$ | $|s|$ | n/a | n/a |
| | # supervised NN nodes | $\{0, 64\}$ | 0 | 0 | n/a | n/a |
| $k$nn | $k$ | $\{1,2,20,100\}$ | 2 | 100 | n/a | n/a |
| DNC | $d$ | $\{1,2,5,10\}$ | 1 | 1 | 10 | 10 |
| | $k$ | $\{10\%,50\%\}$ | 10% | 10% | 10% | 10% |
| | $c$ | $\{10\%,25\%,50\%,75\%\}$ | 25% | 25% | 10% | 10% |

## G  COMPUTATIONAL RESOURCES

Our experiments are conducted on a high-performance cluster with 2.6Ghz CPUs with 56 threads and 64gb RAM per node. The algorithms are coded in Python 3 and we use PyTorch to construct neural network architectures (Paszke et al., 2019). For the maze environment, having relatively long episodes, the average training time was 21 CPU hours. The inventory and job-shop environments took on average 12 and 10 CPU hours to train, respectively.

## H  COMPLEMENTARY RESULTS

In this section, we provide complementary results that support the statements in the main body of the text. First, we present the memory usage and step times of the studied methods over different problem sizes. Next, we show additional convergence curves and boxplots for the maze environment which clarify the performance difference between the various methods. We conducted an additional robustness study with randomized initial agent positions for the maze environment, show evidence of MinMax's and DNC w/o SA's convergence below DNC's average performance in the job-shop scheduling problem, and provide exploration heatmaps for the maze environment.

**Memory consumption**  DNC exhibits a consistent memory usage, even as the complexity of tasks increases, which is beneficial in real-world implementations with limited hardware. Our results, presented in Table 3, show that while the memory requirements for kNN, LAR, and VAC rise substantially when scaling from a maze of size $2^{12}$ (maze with 12 actuators) to $10^{10}$ (job-shop scheduling problem), the memory consumption for DNC remains stable at 68 to 73 MB. The memory consumption in the case of kNN, LAR, and VAC is driven by the need to set up a matrix containing all discrete actions. For example, in the case of the job-shop scheduling problem this would be a matrix with 5 rows and $10^{10}$ columns. We obtained the consumed memory using the Python profiler guppy.

**Algorithm step time**  Figure 7 depicts each algorithm's step time, i.e., the average duration of obtaining a decision from the policy and applying it to the environment, within an episode during

Table 3: Required MB heap consumed per algorithm for the the maze and job-shop environment. The action space size is displayed in parentheses in the top row.

| Algorithm | Maze ($2^{12}$) | Maze ($2^{20}$) | Job-shop ($10^{10}$) |
|---|---|---|---|
| DNC | 68 MB | 68 MB | 73 MB |
| DNC w/o SA | 68 MB | 68 MB | 73 MB |
| MinMax | 68 MB | 68 MB | 73 MB |
| kNN | 68 MB | 252 MB | 373,000 MB |
| LAR | 68 MB | 252 MB | 373,000 MB |
| VAC | 68 MB | 252 MB | 373,000 MB |

testing. As we apply DNC during both testing and training, the reported step times also apply during training. Our findings suggest that while SA increases training time, its benefits in terms of search quality may outweigh these computational costs, especially if the action space is too large for other neighborhood-based methods to (i) learn at all due to memory restrictions (kNN, LAR) or (ii) learn less performant policies (MinMax, DNC w/o SA) due to inaccurate mappings. During training, longer running times are acceptable as the focus is on model accuracy and learning, not speed. In testing and practical implementation, DNC offers step times in the range of milliseconds to a few hundredths of a second, hence remaining practical for real-time decision-making. Note that step time metrics depend on system specifications and may vary with different hardware and software efficiencies.

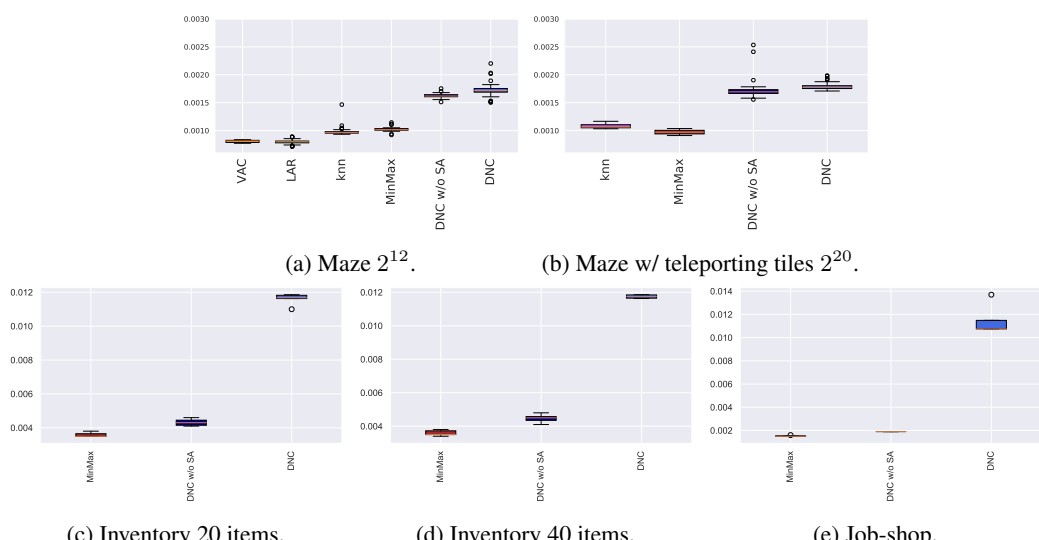

(a) Maze $2^{12}$.  (b) Maze w/ teleporting tiles $2^{20}$.

(c) Inventory 20 items.  (d) Inventory 40 items.  (e) Job-shop.

Figure 7: Average step time in seconds in the maze, inventory, and job-shop environment over 50 seeds during testing.

**Additional convergence curve**   To further illustrate the results in the maze environment with teleporting tiles, we provide an additional graph that shows the convergence curves excluding DNC w/o SA in Figure 8. As can be seen, the confidence intervals of DNC and MinMax overlap only slightly, with DNC converging to a considerably more narrow variance corridor.

**Testing performance for the maze environment**   Since the confidence intervals for the maze domains overlap, it is difficult to clearly observe performance differences. Therefore, we show boxplots of the total expected returns for the final learned policy, i.e., the policy found in the last training episode, in Figure 9. If the agent finds the goal, the episode's total reward is close to 100. If the agent does not find the goal, it is -7.45, due to the small negative step rewards collected over an episode. Consider, for example, DNC in Figure 9a, wherein the agent finds the goal in 49 out of 50 seeds and in one seed the agent fails to do so. The failed seed is an outlier. This example explains the large confidence intervals as shown in the convergence curves. In this example, the standard deviation already amounts to $\approx 15$, even though there is only one outlier.

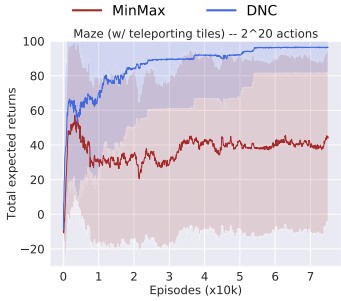

Figure 8: Average total returns during testing for 50 random seeds over the number of training iterations in the maze environment with teleporting tiles for DNC and MinMax. The shaded area represents the 2 standard deviation training seed variance corridor.

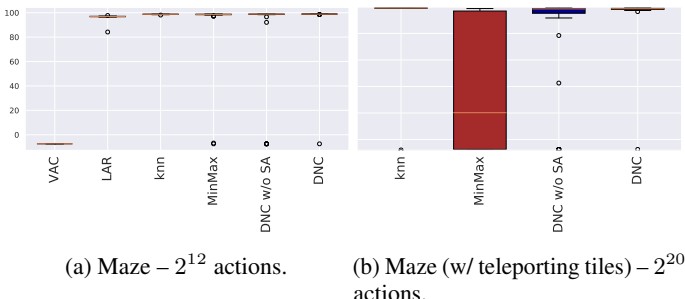

(a) Maze – $2^{12}$ actions.

(b) Maze (w/ teleporting tiles) – $2^{20}$ actions.

Figure 9: Total expected returns during testing for 50 random seeds in the final episode.

**Maze environment with randomized initial positions**    To test the robustness of learned policies, we conducted further testing of the final policy in a maze environment featuring random starting positions. To this end, we used the policies obtained for each seed on the original environment (with a fixed initial position) and executed these policies over 100 episodes with random initial positions. We took the average performance of these 100 episodes and again averaged over the random seeds. The resulting outcomes, detailed in Figure 10, show no significant deviation from earlier results in terms of performance and variance over seeds, implying that DNC learned a robust policy.

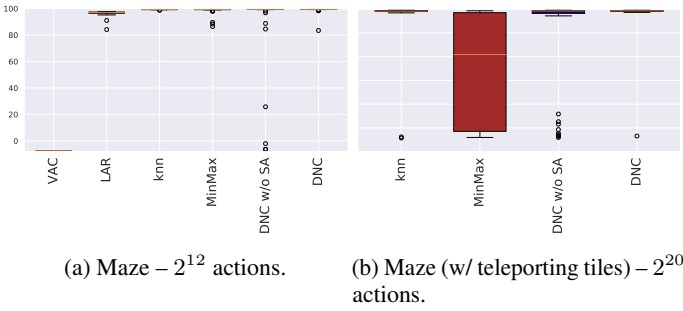

(a) Maze – $2^{12}$ actions.

(b) Maze (w/ teleporting tiles) – $2^{20}$ actions.

Figure 10: Total returns during testing for 50 random seeds in the final episode, using random initial positions.

**Convergence for MinMax and DNC w/o SA in the job-shop scheduling problem**    In Figure 11 we report the convergence graphs of MinMax's and DNC w/o SA's average performance in the job-shop scheduling problem over 80k episodes to provide evidence of their convergence below DNC's average performance.

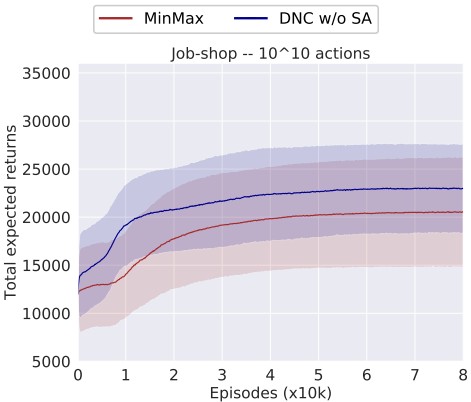

Figure 11: Average total expected returns during testing for 50 random seeds over the training iterations in the job-shop scheduling problem, MinMax and DNC w/o SA only. The shaded area represents the 2 standard deviation training seed variance corridor.

**Maze heatmaps**   We visually compare policies for the standard maze environment. Figure 12 shows exploration heatmaps for all methods (excluding the ablation) for the 12 actuator setting of the maze environment without teleporting tiles. Here, the more frequently a location is visited, the brighter the color, i.e., from least visits to most visits: black-red-orange-yellow. We note that at the start of training, the movement of the agent is still random. However, as the policies converge, a clearer path becomes visible, given the $10\%$ action noise. VAC is unable to handle the large action space and gets stuck in the lower left corner. $k$nn and MinMax eventually find a comparable policy that reaches the goal state, albeit the shortest path is not found. It seems that both policies learned to stay far away from the wall, as hitting the wall often results in getting stuck. Both LAR and DNC found a policy that tracks more closely around the wall, hence taking a shorter path to the goal state. However, LAR got stuck at the lower boundary of the grid for a number of episodes, whereas DNC displayed more productive learning behavior.

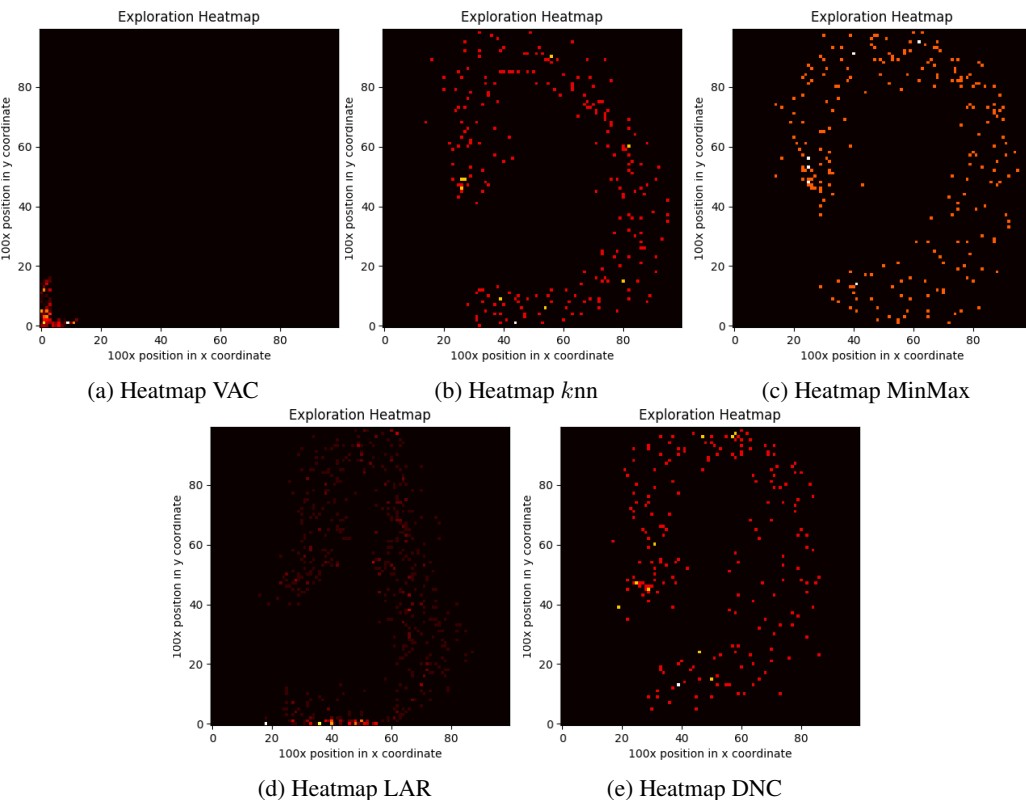

(a) Heatmap VAC

(b) Heatmap $k$nn

(c) Heatmap MinMax

(d) Heatmap LAR

(e) Heatmap DNC

Figure 12: Exploration heatmap over all training episodes of each method for the 12 actuator case.

