# OpenReview forum: "Dynamic Neighborhood Construction for Structured Large Discrete Action Spaces"
_ICLR.cc/2024/Conference — ICLR 2024 poster_

### Official Review · Reviewer_kH6M · 2023-10-30

**Soundness:** 3 good
**Presentation:** 4 excellent
**Contribution:** 3 good
**Rating:** 8
**Confidence:** 2

**Summary:**

The authors present an approach for learning in environments with large, discrete action spaces. Specifically, they learn policies in a continuous space that can be mapped to the discrete actions. The mapping the propose leverages structure in the discrete space in which it performs a localized search to efficiently perform the mapping.

**Strengths:**

**Originality:** The work presented in this paper is novel as far as I know. They propose an approach that has not been done before that attempts to mitigates drawbacks exhibited by other algorithms.

**Quality:** The problem is well-motivated and the drawbacks of current SOTA approaches for the problem are identified and the authors describe sufficiently how their approach attempts to tackle one of these issues. While I have not delved thoroughly into the appendix to check every bit of the theory, the sketched out lemmas seem sound. The experiments conducted were reasonable and thorough enough (I appreciate the ablation study performed.)

**Clarity:** Easy to read and understand.

**Significance:** I believe this work could be impactful to others in the area.

**Weaknesses:**

My one issue is with the first set of experiments in the maze. The results do not really demonstrate that DNC actually provides any advantage over the other baselines. While the second set of experiments do, I think maybe adding another domain may support performance claims by the authors. Alternatively, if the argument is that the performance of DNC is competitive to other SOTA, what other benefits may it provide (better wall-clock time, memory usage, etc.)?

**Questions:**

Tiny nitpick: The second sentence in Problem Definition --- it would read better to describe the states and actions in the order you introduce them.

---

> ### Author Response · Authors · 2023-11-22
> **Thank you for your comments!**
>
> Thank you for providing highly useful feedback and valuing our paper’s originality, the quality of our
> proposed approach, the clarity of our write-up, and our paper’s significance.
>
> **Weakness:** We agree that the maze environment is not ideal to benchmark our DNC method, as also mentioned by the other reviewers. However, we included the maze environment as it was used like this in prior works on large discrete action spaces [cf. 1, 2]. Accordingly, we decided to study the same environment to contextualize our results. Its purpose is to show that our approach does not fall behind the quality of other approaches in established environments.
>
> The motivation for DNC are contextual multi-stage stochastic optimization problems that arise in real-world applications. We followed your suggestion and added another environment from those problem domains, specifically a dynamic job-shop scheduling problem [cf. 3, 4]. In this  extensively studied problem a set of $G$ jobs needs to be allocated to $N$ machines that can serve up to $L$ jobs each, yielding an action space of size $|\mathcal{A}|=L^N$. We consider a load-balancing variant of this problem, for which jobs need to be balanced over heterogeneous machines. The action space in our experimental design amounts to $10^{10}$ actions, hence only DNC, DNC w/o SA, and MinMax can learn policies at all. The results show that DNC significantly outperforms DNC w/o SA and MinMax. For a more thorough results discussion as well as the updated convergence graphs in Figure 3, we refer to the updated paper. To keep the results section concise, we replaced the maze environment with 28 actuators by the job-shop scheduling results.
>
> Moreover, we improved the manuscript according to your suggestions and now discuss DNC's memory advantage. DNC exhibits a consistent memory usage even as the action space size increases, highlighting its practical benefits. Our results, presented in Table 3 in Appendix H (see also pasted below), show that while the memory requirements for kNN, LAR, and VAC rise substantially when scaling from a maze of size $2^{12}$ to a job-shop environment with $10^{10}$ actions, the memory consumption for DNC remains stable at 68 to 73 MB. The memory consumption in the case of kNN, LAR, and VAC is driven by the need to setup a matrix containing all discrete actions. For example, in the case of the job-shop scheduling problem this would be a matrix with 5 rows and $10^{10}$ columns. We obtained the consumed memory using the Python profiler guppy. The stability of DNC is vital for applications with limited resources or when scaling for larger datasets without a significant increase in resource demands. Its constant memory footprint allows it to handle larger action spaces efficiently without performance loss due to memory constraints. This scalability is critical in real-world applications with finite computational resources. DNC's ability to maintain consistent memory usage, regardless of the action space size, makes it an essential tool in machine learning, particularly when resources are limited.
>
> | **Algorithm** | Maze ($2^{12}$) | Maze ($2^{20}$) | Job-shop ($10^{10}$) |
> |--------------------|-------------------|------------------|----------------------|
> | DNC               | 68 MB           | 68 MB           |               73 MB |
> | DNC w/o SA   | 68 MB           | 68 MB           |               73 MB |
> | MinMax          | 68 MB           | 68 MB           |               73 MB |
> | kNN                | 68 MB           | 252 MB          |     373,000 MB |
> | LAR                | 68 MB           | 252 MB          |     373,000 MB |
> | VAC                | 68 MB           | 252 MB          |     373,000 MB |
>
>
>
> **Questions:** Thank you for the suggestion. We have altered the order in this sentence.
>
> If you are satisfied with our improvements, we would kindly ask you to consider raising the score for
> our manuscript.

---

> > ### Comment · Reviewer_kH6M · 2023-11-22
> >
> > I thank the authors for their responses and additions to the paper. I believe it will make the paper a lot stronger. My score, though, will remain the same.

---

> > > ### Author Response · Authors · 2023-11-23
> > >
> > > Thank you for your kind words and for recognizing the efforts we've put into strengthening the paper. Your feedback is greatly appreciated.

---

### Official Review · Reviewer_x32f · 2023-10-31

**Soundness:** 3 good
**Presentation:** 3 good
**Contribution:** 3 good
**Rating:** 6
**Confidence:** 2

**Summary:**

Given the need in fields like inventory management, an efficient method for RL with structured large discrete action spaces (n-dimensional lattices) is proposed. First a continuous action is generated and discretized, then through simulated annealing with perturbation defined by other actions on the lattice with low hamming distance, a final proposed discrete action is obtained. This avoids the necessity of constructing extremely large nearest neighbor graphs, or solving a linear program on every forward pass. The convergence for the action selection process is proven, and the method is evaluated in two simulated domains with strong results over baselines.

**Strengths:**

- The method is well-motivated
- Results are strong over baselines in Figure 3
- Components are ablated

**Weaknesses:**

I should first couch my review by noting I'm not the most familiar with the literature in this area, so this is an educated guess:

- The maze domain seems a bit too trivial to be useful, it is informative for a demonstration of the method, but unlike the inventory management environment, I'm not sure what the real-world justification might be. I would prefer to have at least one more grounded and complex environment (perhaps a video game with many different possible actions, perhaps a multi-agent setting? I'm not the most familiar with the domains in this subfield).
- Perhaps this is my unfamiliarity with the domain, but sometimes the text is quite hard to follow. I think part of this is the reliance on lots of acronyms (LDAS, SLDAS, MILP, SA), but also the claim in Section 2 "we formulate the task of finding a Q-value maximizing a as a mixed-integer linear program" was a bit confusing on first read, as really the authors are proposing an approximate solution, so saying this upfront might be a bit more clear.
- Would be nice to see MILP results in Figure 5 against the proposed method and other baselines in Figure 3, and the discussion at the end of Appendix B is informative and it may be nice to have a note in the text

**Questions:**

See Weaknesses

---

> ### Author Response · Authors · 2023-11-22
> **Thank you for your comments**
>
> Thank you for the valuable feedback. We appreciate the compliments on our paper's motivation and results. Please find our detailed answers to all questions and comments below.
>
> **W1:**
> We appreciate your feedback regarding the choice of the maze domain and the suggestion for more complex environments. We included the maze environment as it was used like this in prior works on large discrete action spaces [1,2]. Accordingly, we decided to study the same environment to contextualize our results. Its purpose is to show that our approach does not fall behind the quality of other approaches in established environments.
>
> The motivation for DNC are contextual multi-stage stochastic optimization problems that arise in real-world applications. We followed your suggestion and added another environment from those problem domains, specifically a dynamic job-shop scheduling problem [cf. 3,4]. In this  extensively studied problem a set of $G$ jobs needs to be allocated to $N$ machines, that can serve up to $L$ jobs each, yielding an action space of size $|\mathcal{A}|=L^N$. We consider a load-balancing variant of this problem, for which jobs need to be balanced over heterogeneous machines. The action space in our experimental design amounts to $10^{10}$ actions, hence only DNC, DNC w/o SA, and MinMax can learn policies at all. The results show that DNC significantly outperforms DNC w/o SA and MinMax. For a more thorough results discussion as well as the updated convergence graphs in Figure 3, we refer to the updated paper. To keep the results section concise, we replaced the maze environment with 28 actuators by the job-shop scheduling results.
>
> **W2:**
> Thank you for your suggestions. We understand that the abbreviations, while aimed at maintaining conciseness to adhere to the page limit, may have impacted the readability to some extent. Because of the page limit, we decided to keep the abbreviations in the main document. We made several textual changes to enhance the clarity of the paper.
>
> We have improved the sentence related to the MILP, highlighting that we only use the MILP to formalize the problem but aim for an efficient algorithm to solve it.
>
> **W3:**
> Thank you for pointing this out. We added the remaining results in Figure 5 and added a note referring to Appendix B in the main text.
>
> If you are satisfied with our improvements, we would kindly ask you to consider raising the score for our manuscript.

---

> ### Comment · Reviewer_x32f · 2023-11-22
>
> Thanks for your response, I realize some of my phrasing was quite poor on Weakness 3: I wanted the results for MILP and Hybrid to be presented in Figure 3 in the main text, not for results on many baselines from Figure 3 to be added to Figure 5. Barring this change, all of my concerns are answered and I would be happy to raise my score, pending discussion with other reviewers.

---

> > ### Author Response · Authors · 2023-11-23
> >
> > Thank you for the fast reply, we highly value your feedback on our paper. We appreciate your consideration of our improvements and apologize for the misunderstanding. We would be very grateful if you could consider increasing your score after discussion with the other reviewers.

---

### Official Review · Reviewer_ZSBT · 2023-11-01

**Soundness:** 3 good
**Presentation:** 3 good
**Contribution:** 3 good
**Rating:** 8
**Confidence:** 4

**Summary:**

This paper addresses the problem of reinforcement learning in large discrete action spaces, which vanilla techniques typically cannot handle, and require adaptations. The authors propose a technique that, similarly to other works, outputs a continuous action, which is then mapped to a valid discrete action that is executed in the environment. Their proposal applies for action spaces that are regularly structured (e.g., an equally-spaced grid), which allows the method to circumvent needing to store the valid actions in memory. Instead, to map to a discrete action, a simulated annealing search is proposed, that is shown (via reasonable assumption and a standard analysis technique) to yield actions that improve the (estimated) Q-value. The method is evaluated in environments with an exponential number of actions, showing similar performance with prior methods in regimes with less actions, and better performance in regimes with more actions.

**Strengths:**

**S1**. The proposed technique is simple, elegant, and rigorously analyzed. It is also shown to work well empirically in environments that satisfy the required conditions.

**S2**. The writing of the paper is excellent and clear.

**Weaknesses:**

**W1**. My primary criticism is that the considered environments and scenarios where the technique applies are somewhat contrived. If one is faced with such a huge decision space, we would typically expect to spend the time in the modelling stage of the problem to design an action space that is substantially more manageable. For example, in the maze environment, we could design a discrete set of actions (move left, etc.), with the actual execution being handled by lower-level control primitives. Indeed, this needs to be performed for each individual problem (as the authors do mention).

**W2**. I also have some concerns about the evaluation:

- Firstly, for the maze environments, the variability of the techniques is such that the confidence intervals overlap substantially, and it is not possible to draw reliable conclusions from these figures alone. Would this still happen with substantially more seeds (e.g. 50+)? Alternatively, would another procedure (e.g., running many episodes with randomly initialised positions) yield more reliable (i.e., with non-overlapping CIs)?
- The simulated annealing search procedure must introduce a computational overhead compared to picking the best action greedily (e.g., as in DNC w/o SA). This is not mentioned, and the impact on runtimes is not analyzed or discussed. This needs to be addressed, accompanied by measurements.

**Questions:**

**C1**. The paper refers in several places to approaches for "unstructured" LDAS (abstract, intro, figure 1, etc.). This is somewhat of a misnomer since these problems *do* exhibit structure (i.e., the action space is such that actions that are close in embedding space yield similar Q-values and lead to similar outcomes when actuated in the environment). I think a better term would be *irregularly* structured. Similarly, the proposed method applies to *regularly* structured action spaces.

**C2**. Related to the second point of W2 above, the paper mentions the use of "SA to efficiently search across different and potentially better neighborhoods" (Discussion on p6). While this statement potentially applies for the memory efficiency, it certainly doesn't in terms of time, and should be qualified. SA in general is not computationally efficient.

**C3**. Small nitpick: "Constraints 2" -> "Constraint 2", etc. in the text at the end of Section 2.

---

> ### Author Response · Authors · 2023-11-22
> **Thanks for your comments**
>
> Thank you for providing valuable feedback and appreciating our paper's writing, the simplicity of our approach, and our rigorous analysis. Below we answer to all questions and comments.
>
> **W1:** We agree that the huge action space in the maze could be avoided by refined modeling. We included the maze as it was used like this in prior works on large discrete action spaces [1,2]. Accordingly, we decided to study the same environment to contextualize our results. Its purpose is to show that our approach does not fall behind the quality of other approaches in established environments.
>
> We believe using DNC and a refined modeling effort are not necessarily mutually exclusive. For example, one can use DNC to create benchmarks to verify the performance of the modeling approach. The main motivation of our work does not relate to problems handled by refined modeling, but stems from contextual stochastic multi-stage optimization settings grounded in real-world applications. Such settings, e.g., the inventory and job-shop environments, often lack room for action space simplification, as it would infer a significant loss in solution quality.
>
> **W2.1:** We re-ran the maze with 12 actuators and with teleporting tiles (50 seeds). For statistical significance and consistency we also ran 50 seeds for the other environments. To reduce the computational burden and save space in the main text, we excluded the results on the maze with 28 actuators. We report the results in Fig 3.
>
> * For the maze with 12 actuators, CIs are narrower than in the original submission. However, CIs still overlap and the avg. performances of the different approaches do not differ significantly from the results in the original paper.
>
> * For the maze with teleporting tiles, we reconfirm previous observations. The CI of DNC overlaps with MinMax's CI only slightly, showing DNC's advantage over MinMax in this environment. DNC is therefore able to provide high quality policies in more complex environments, comparable to kNN. This result is noteworthy, as MinMax is the only SOTA method able to scale to very large SLDAS such as the inventory environment. To disclose the CI overlap of DNC and MinMax, we included Fig 8 in Ap. H and summarized the findings in the main paper.
>
> In addition, we added boxplots in the appendix, displaying the avg. performance across the maze in the final training episode (see Fig 9 in Ap. H) and summarized the findings in the main paper. The results suggest that convergence in the maze is much more stable than perceived (cf. Fig 3). The wide CIs can be attributed to the inherent reward variance of the environment. We refer to Ap. H, Fig 10 for a more thorough discussion.
>
> We addressed your alternative suggestion to run several episodes with random initial positions. The resulting outcomes and discussion are detailed in Fig 11 in Ap. H.
>
> **W2.2:** We measured the avg. duration of obtaining a decision from the policy and applying it to the environment, i.e., step time, within an episode during testing. We apply DNC during both testing and training, so the reported step times also apply during training.  We have added a discussion and Fig 7 in Ap. H, wherein we compare avg. step times (50 seeds) for the benchmarks. We added a reference in the main paper.  Our findings suggest that while SA increases training time, its benefit in terms of search quality may outweigh these costs, especially if the action space is so large that other methods fail to learn at all due to memory restrictions (kNN) or learn less performant policies (MinMax, DNC w/o SA). During training, we consider longer computational times to be acceptable as the focus is on model accuracy, not speed. In practical implementation, DNC offers step times in the range of milliseconds, remaining practical for real-time decision-making.
>
> **C1:** We added the proposed terms and made a clarification in the beginning of paragraph 2. In the rest of the paper we kept the original submission's "structured" and "unstructured" for conciseness.
>
> **C2:** We have adjusted wording on page 6 to more accurately reflect the characteristics of SA. SA's time impact within DNC remains controllable, see Fig 7 in Ap. H. The max. no. SA iterations is limited by DNC's hyperparameters. The max. no. SA search iterations needed to achieve good performance is limited to 10 SA search iterations in the inventory environment (cf. Table 2, Ap. F). During a single SA iteration, DNC evaluates multiple neighbors in parallel through our dynamic neighborhood construction procedure, see Section 3. Hence, evaluating a larger neighborhood does not necessarily require more SA search iterations.
>
> **C3:** Thank you for the comment. We use the plural to emphasize that we refer to a set of constraints. For example, Equation (4) has to hold for all neurons $d\in \mathcal{D}_2$.
>
> If you are satisfied with our improvements, we would kindly ask you to consider raising the score for our manuscript.

---

> > ### Comment · Reviewer_ZSBT · 2023-11-23
> > **Post-rebuttal response to authors**
> >
> > Thanks for engaging with my suggestions! I think the additional results have substantially improved the robustness of the findings. Given this, I am increasing my score to an 8.

---

> > > ### Author Response · Authors · 2023-11-23
> > >
> > > Thank you for your quick response. Your appreciation of the additional effort we've invested in enhancing the paper is important to us, and we highly value your feedback.

---

### Author Response · Authors · 2023-11-22
**Global response**

Thank you for the insightful and positive feedback. We have addressed each remark of every reviewer, including clarifying our choices of environments, adding a new environment, and enriching numerical studies and discussions. We provide a brief summary of changes in the paper below:

1. We addressed reviewer x32f's and kH6M's wish to incorporate an additional real-world inspired domain, the job-shop scheduling problem, which confirmed DNC's superiority over the benchmark methods, hence strengthening our numerical evaluation (cf. Figure 3 and discussion in Section 5). Due to the page limit and to keep the discussion concise, we shortened the discussion on the maze environment by replacing the results and discussion regarding the maze environment with 28 actuators with the new results of the job-shop scheduling problem.

2. As per reviewer ZSBT's suggestion we expanded our experiments by considering 40 additional seeds for all environments, such that we now report results over 50 random seeds. The additional seeds demonstrate the robustness of DNC.

3. We improved wording and various sentences in Section 2 following all reviewers' suggestions to enhance clarity and readability.

4. We have responded to specific feedback from Reviewer ZSBT on modeling in complex action spaces. In this context, also based on Reviewer x32f's comment, we clarified the motivation of the maze domain in Section 4.

5. We added insights considering step times of the various methods, to quantify SA's impact on computational effort, as suggested by reviewer ZSBT (see, e.g., Appendix H, Figure 7).

6. Based on reviewer kH6M's comments, we added further experiments regarding the different methods' memory usage, emphasizing DNC's advantages in terms of low memory usage compared to VAC, kNN, and LAR (Appendix H, Table 3).

7. We conducted additional experiments with randomized initial agent positions, as suggested by reviewer ZSBT (Appendix H, Figure 10).

8. We added the remaining results to the MILP result graph in the appendix, as suggested by reviewer x32f (Appendix B, Figure 5).

All additional experiments that have been mentioned here with references to detailed results in appendices have also been reflected in the discussion in the main body of the paper in an aggregated form.

The revised manuscript reflects these changes, offering a more comprehensive discussion of our results. We are grateful for your guidance, which helped to significantly improve our work. If you are satisfied with our improvements, we kindly ask to consider raising the score for our manuscript.

To avoid repeating references in the individual reviewer rebuttals, we added the reference list to this global response.

**References**

[1] Yash Chandak, Georgios Theocharous, James Kostas, Scott Jordan, and Philip Thomas. Learning
action representations for reinforcement learning. In International Conference on Machine Learning,
pages 941–950. PMLR, 2019.

[2] Yash Chandak, Georgios Theocharous, Chris Nota, and Philip S. Thomas. Lifelong learning with a
changing action set. In AAAI Conference on Artificial Intelligence, 2019.

[3] Xinquan Wu and Xuefeng Yan. A spatial pyramid pooling-based deep reinforcement learning model
for dynamic job-shop scheduling problem. Computers \& Operations Research, 160:106401, 2023

[4] Cong Zhang, Wen Song, Zhiguang Cao, Jie Zhang, Puay Siew Tan, and Xu Chi. Learning to dispatch
for job-shop scheduling via deep reinforcement learning. In H. Larochelle, M. Ranzato, R. Hadsell,
M.F. Balcan, and H. Lin, editors, Advances in Neural Information Processing Systems, volume 33,
pages 1621–1632. Curran Associates, Inc., 2020.

---

### Meta-Review · Area_Chair_1vqr · 2023-12-05

**Metareview:**

(a) this work aims to tackle the problem of RL in large discrete state-action spaces like warehouse operations or drug design. The key idea they use is to convert continuous actions into discrete ones and leveraging dynamic neighborhoods to just search locally in the discrete action space. This allows for the usage of problem structure, and can be embedded into actor-critic methods. They provide theory and simple experiments to back this up.

(b) the problem setting is immensely useful, and the proposed method is both practical and has provable performance.  The empirical results do show a big benefit on the considered domains.

(c) the primary issue was with the experimental domains being either very simple or somewhat contrived. Better experimental evaluation would go a long way with this paper to make it stand out more.

(d) Further experimental validation, clarification on terminology and jargon throughout the paper.

**Justification For Why Not Higher Score:**

The experimental validation is not compelx enough and the theory is not comprehensive enough for a higher score.

**Justification For Why Not Lower Score:**

The idea seems sound, and well described with both theory and some experiments. This seems like a useful idea to present to the community.

---

### Decision · Program_Chairs · 2024-01-16

Accept (poster)